# Efficient Sparse-Reward Goal-Conditioned Reinforcement Learning with a High Replay Ratio and Regularization

## Abstract

Reinforcement learning (RL) methods with a high replay ratio (RR) and regularization have gained interest due to their superior sample efficiency. However, these methods have mainly been developed for dense-reward tasks. In this paper, we aim to extend these RL methods to sparse-reward goal-conditioned tasks. We use Randomized Ensemble Double Q-learning (REDQ) (Chen et al., 2021), an RL method with a high RR and regularization. To apply REDQ to sparse-reward goal-conditioned tasks, we make the following modifications to it: (i) using hindsight experience replay and (ii) bounding target Q-values. We evaluate REDQ with these modifications on 12 sparse-reward goal-conditioned tasks of Robotics (Plappert et al., 2018), and show that it achieves about $2\times$ better sample efficiency than previous state-of-the-art (SoTA) RL methods. Furthermore, we reconsider the necessity of specific components of REDQ and simplify it by removing unnecessary ones. The simplified REDQ with our modifications achieves $\sim 8\times$ better sample efficiency than the SoTA methods in 4 Fetch tasks of Robotics.

## 1 Introduction

In the reinforcement learning (RL) community, improving the sample efficiency of RL methods has been important. RL methods have been promising for solving complex control tasks, including dexterous in-hand manipulation (Andrychowicz et al., 2020), quadrupedal/bipedal locomotion (Lee et al., 2020; Haarnoja et al., 2023), and car/drone racing (Wurman et al., 2022; Kaufmann et al., 2023). However, RL methods are generally data-hungry and require large amounts of training samples to solve tasks (Mendonca et al., 2019). Motivated by this problem, various sample-efficient RL methods have been proposed (Haarnoja et al., 2018; Lillicrap et al., 2015; Schulman et al., 2017; Fujimoto et al., 2018).

In recent years, RL methods using a high replay ratio (RR) and regularization have attracted attention as sample-efficient methods (Janner et al., 2019; Chen et al., 2021; Hiraoka et al., 2022; Nikishin et al., 2022; Li et al., 2023a; D'Oro et al., 2023; Smith et al., 2023b; Sokar et al., 2023; Schwarzer et al., 2023). RR is the ratio of components (e.g., policy and Q-functions) updates to the actual interactions with an environment. A high RR facilitates sufficient training of the components within a few interactions but exacerbates the components' overfitting. Regularization techniques (e.g., ensemble (Chen et al., 2021) or dropout (Hiraoka et al., 2022)) are employed to prevent the overfitting. The RL methods equipped with them have exhibited high sample efficiency and enabled training agents within mere tens of minutes in real-world tasks, such as quadrupedal robot locomotion (Smith et al., 2022; 2023a), robotic manipulation (Luo et al., 2024), and image-based vehicle driving (Stachowicz et al., 2023).

However, these methods have been developed mainly on dense-reward tasks rather than sparse-reward tasks. Many RL tasks require RL methods to learn with a sparse reward due to the difficulty of designing dense rewards (Andrychowicz et al., 2017; Trott et al., 2019; Agrawal, 2022; Knox et al., 2023; Booth et al., 2023). A typical example of such tasks is **sparse-reward goal-conditioned tasks** (Plappert et al., 2018), where a positive reward is provided only upon successful goal attainment. RL methods that can efficiently learn in these tasks hold substantial value in numerous application scenarios, such as (i) developing versatile agents capable of achieving diverse goals (Vithayathil Varghese & Mahmoud, 2020; Beck et al., 2023), or (ii) con-

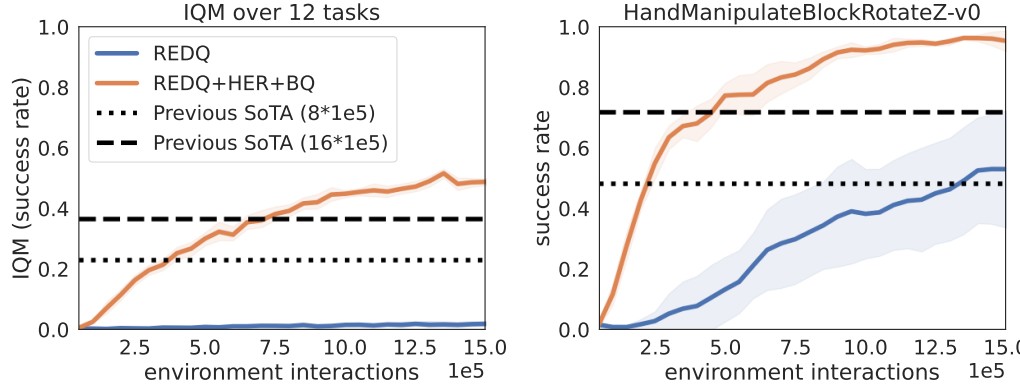

Figure 1: The task success rate of vanilla REDQ and our modified REDQ (REDQ+HER+BQ). The left-hand side figure shows the interquartile mean (IQM) with a 95% confidence interval (Agarwal et al., 2021) for the success rate over 12 Robotics tasks. The right-hand side figure shows the average scores with one standard deviation in the HandManipulateBlockRotateZ task (one of the Robotics tasks). We also present scores from previous SoTA methods with $8 \cdot 10^5$ and $16 \cdot 10^5$ samples (number of environment interactions). For context, $10^5$ samples correspond to approximately one hour of real-world experience. The left-hand side figure shows that our modified REDQ achieves approximately $2\times$ better sample efficiency than previous SoTA methods. Examples of policies learned by our modified REDQ are showcased in the video in our supplementary file.

structing low-level agents to execute goals provided by high-level agents in a hierarchical framework (Pateria et al., 2021; Brohan et al., 2023; Yu et al., 2023). Therefore, it is valuable to investigate whether RL methods with a high RR and regularization efficiently work in sparse-reward goal-conditioned tasks.

In this paper, we apply an RL method with a high RR and regularization to sparse-reward goal-conditioned tasks. As our sparse-reward goal-conditioned tasks, we consider Robotics (Plappert et al., 2018) (Section 2). As an RL method with a high RR and regularization, we employ Randomized Ensemble Double Q-learning (REDQ) (Chen et al., 2021) (Section 3.1). To adapt REDQ for the Robotics tasks, we introduce the following modifications to REDQ: (i) using hindsight experience replay (HER; Section 3.2) and (ii) bounding target Q-value (BQ; Section 3.3). We experimentally demonstrate that REDQ with these modifications can achieve better sample efficiency than previous state-of-the-art (SoTA) RL methods (Fig. 1. See Sections 4 and 5 for more comprehensive details).

While our main contribution is **successful application of the RL method with a high RR and regularization to sparse-reward goal-conditioned tasks**, we make two additional significant contributions:
**1. Provision of insights on BQ in HER usage:** Previous works on HER (e.g., Andrychowicz et al. (2017); Zhao & Tresp (2018; 2019); Xu et al. (2023)) applied BQ to their base RL method (a deep deterministic policy gradient; DDPG (Lillicrap et al., 2015)). However, these works did not investigate (i) the contribution of BQ to performance improvements, (ii) the underlying rationale for its use, and (iii) its effectiveness beyond DDPG [1]. We (i) conducted ablation studies for BQ and revealed its contribution on performance improvements (Figs. 5 and 11), (ii) empirically demonstrated its rationale from the perspective of Q-function stability (Figs. 4 and 12), and (iii) showed its effectiveness for REDQ (Chen et al., 2021) and Reset (Nikishin et al., 2022) (Figs. 5 and 11).
**2. Simplification of REDQ in sparse-reward goal-conditioned tasks:** REDQ uses clipped double Q-learning and an entropy term in its target Q-value calculation. We find that REDQ can be simplified by removing them (Figs. 8 and 9 in Section 5). Remarkably, the simplified REDQ with our modifications achieves $\sim 8\times$ better sample efficiency than SoTA methods in the Fetch tasks of Robotics (Fig. 9). Our findings may be valuable in maintaining the simplicity of REDQ, which improves reproducibility and reduces human effort in debugging and engineering.

---

[1]See "Bounding Q-value" in Section 6 for more detailed discussion

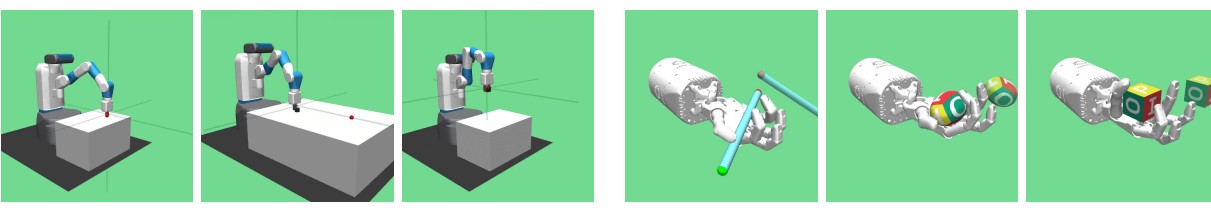

(a) Fetch tasks                                         (b) HandManipulate tasks

Figure 2: Robotics (Plappert et al., 2018) tasks.

## 2    Preliminary: Sparse-Reward Goal-Conditioned RL

We focus on sparse-reward goal-conditioned RL. This is typically modeled as goal-augmented Markov decision processes $\langle \mathcal{S}, \mathcal{A}, \mathcal{G}, \gamma, p_{s_0}, p_g, \mathcal{T}, \mathcal{R} \rangle$ (Liu et al., 2022) with sparse rewards. Here, $\mathcal{S}$, $\mathcal{A}$, $\mathcal{G}$, and $\gamma$ are the state space, the action space, the goal space, and the discount factor, respectively. $p_{s_0} : \mathcal{S} \to [0,1]$ is the initial state distribution. $p_g : \mathcal{G} \to [0,1]$ is the goal distribution. $\mathcal{T} : \mathcal{S} \times \mathcal{A} \times \mathcal{S} \to [0,1]$ is the dynamics transition function. $\mathcal{R} : \mathcal{S} \times \mathcal{A} \times \mathcal{G} \to \mathbb{R}$ is the reward function, which is sparsely structured. At the beginning of an episode, an agent receives the desired goal $g \sim p_g(\cdot)$. At each discrete time step $t$, an environment provides the agent with a state $s_t \in \mathcal{S}$, the agent responds by selecting an action $a_t \in \mathcal{A}$, and then the environment provides the next reward $r_t \leftarrow \mathcal{R}(s_t, a_t, g)$ and state $s_{t+1} \in \mathcal{S}$. For convenience, as needed, we use the simpler notations of $r$, $s$, $a$, $s'$, and $a'$ to refer to a reward, state, action, next state, and next action, respectively. In addition, as needed, we use the notation of $g'_t \in \mathcal{G}$ and $r'_t \leftarrow \mathcal{R}(s_t, a_t, g'_t)$ to refer to the goal and reward at $t$, respectively. The objective of sparse-reward goal-conditioned RL is to learn a goal-conditioned policy $\pi : \mathcal{S} \times \mathcal{G} \times \mathcal{A} \to [0,1]$ that maximizes the expected cumulative rewards:

$$\mathbb{E}_{a_t, g, s_{t+1}, s_0} \left[ \sum_{t=0}^{\infty} \gamma^t r_t \right], \quad a_t \sim \pi(\cdot|s_t, g), \quad g \sim p_g(\cdot), s_{t+1} \sim \mathcal{T}(\cdot|s_t, a_t), \quad s_0 \sim p_{s_0}(\cdot).$$

As benchmark tasks for sparse-reward goal-conditioned RL, we employ the Robotics (Plappert et al., 2018; de Lazcano et al., 2023) tasks (Fig. 2). In these tasks, an RL agent aims to learn control policies for moving and reorienting objects (e.g., a block or an egg) to target positions and orientations. The reward is sparsely structured: The agent receives a positive reward of 0 if the distance between the positions (and orientations) of the object and the target is within a small threshold, and a negative reward of $-1$ otherwise [2]. We use 12 Robotics tasks for our experiments: FetchReach, FetchPush, FetchSlide, FetchPickAndPlace, HandManipulatePenRotate, HandManipulateEggRotate, HandManipulatePenFull, HandManipulateEggFull, HandManipulateBlockFull, HandManipulateBlockRotateZ, HandManipulateBlockRotateXYZ, and HandManipulateBlockRotateParallel.

## 3    Our RL Method

In this section, we introduce our method for sparse-reward goal-conditioned RL. The algorithmic description of our method is summarized in Algorithm 1. We use REDQ for our base method (Section 3.1). To apply REDQ to sparse-reward goal-conditioned tasks, we make two modifications to REDQ: (i) using hindsight experience replay (HER; Section 3.2) and (ii) bounding target Q-values (BQ; Section 3.3).

### 3.1    Base Method: RL Method with a High RR and Regularization

Our base method is REDQ (Chen et al., 2021), an RL method with a high RR and regularization:
**High RR.** REDQ uses a high RR $G$ (typically $G > 1$), which is the number of Q-function updates (lines 6–12 in Algorithm 1) relative to the number of actual interactions with the environment (line 3). A high RR promotes sufficient training of Q-functions within a few interactions. However, it may cause overfitting of Q-functions and degrade sample efficiency.

---

[2]A more detailed task description can be found at https://robotics.farama.org/.

---

**Algorithm 1** REDQ with our modifications (HER and BQ)

---

Initialize policy parameters $\theta$, $N$ Q-function parameters $\phi_i$, empty replay buffer $\mathcal{D}$, and episode length $T$. Set target parameters $\bar{\phi}_i \leftarrow \phi_i$, for $i = 1, ...., N$.

1: Sample goal $g \sim p_g(\cdot)$ and initial state $s_0 \sim p_{s_0}(\cdot)$

2: **for** $t = 0, .., T$ **do**

3:     Take action $a_t \sim \pi_\theta(\cdot|s_t)$; Observe reward $r_t$ and next state $s_{t+1}$.

4:     **if** $t = T$ **then**

5:        $\mathcal{D} \leftarrow \mathcal{D} \bigcup \{(s_t, a_t, r_t, s_{t+1}, g)\}_{t=0}^T$; Select new goal $g_t'$; Calculate new reward $r_t' \leftarrow \mathcal{R}(s_t, a_t, g_t')$; $\mathcal{D} \leftarrow \mathcal{D} \bigcup \{(s_t, a_t, r_t', s_{t+1}, g_t')\}_{t=0}^T$

6:     **for** $G$ updates **do**

7:        Sample a mini-batch $\mathcal{B} = \{(s, a, r, s', g)\}$ from $\mathcal{D}$.

8:        Sample a set $\mathcal{M}$ of $M$ distinct indices from $\{1, 2, ..., N\}$.

9:        Compute the target Q-value $y$ (same for all $N$ Q-functions):

$$y = r + \gamma \min \left( \max \left( \min_{i \in \mathcal{M}} Q_{\bar{\phi}_i}(s', a', g), Q_{\min} \right), Q_{\max} \right) - \alpha \log \pi_\theta(a'|s', g), \quad a' \sim \pi_\theta(\cdot|s', g)$$

10:     **for** $i = 1, ..., N$ **do**

11:        Update $\phi_i$ with gradient descent using

$$\nabla_\phi \frac{1}{|B|} \sum_{(s,a,r,s',g) \in \mathcal{B}} (Q_{\phi_i}(s, a, g) - y)^2$$

12:        Update target networks with $\bar{\phi}_i \leftarrow \rho\bar{\phi}_i + (1 - \rho)\phi_i$.

13:     Update $\theta$ with gradient ascent using

$$\nabla_\theta \frac{1}{|B|} \sum_{s \in \mathcal{B}} \left( \frac{1}{N} \sum_{i=1}^N Q_{\phi_i}(s, a, g) - \alpha \log \pi_\theta(a|s, g) \right), \quad a \sim \pi_\theta(\cdot|s, g)$$

---

**Regularization.** To mitigate overfitting, our REDQ uses (i) ensemble and (ii) layer normalization. (i) Ensemble of $N$ Q-functions is used as a regularization technique (lines 8–9). Specifically, a random subset $\mathcal{M}$ of the ensemble is selected (line 8) and used for target calculation (line 9). Each Q-function in the ensemble is randomly and independently initialized but updated with the same target (lines 10–11). (ii) Layer normalization (Ba et al., 2016) is applied after the weight layer in each Q-function. Layer normalization is not used in the original REDQ paper (Chen et al., 2021), but its subsequent works (Hiraoka et al., 2022; Ball et al., 2023) show that it further suppresses the overfitting and improves sample efficiency of REDQ. Following these subsequent works, we use layer normalization for our REDQ.

REDQ has demonstrated high sample efficiency in dense-reward continuous-control tasks (Brockman et al., 2016; Fu et al., 2020) based on MuJoCo (Todorov et al., 2012) (see e.g., Chen et al. (2021)) [3]. However, when applied to sparse-reward goal-conditioned tasks, it performs worse than previous SoTA methods (Fig. 1). In the following sections, we will make modifications to improve REDQ's performance in sparse-reward goal-conditioned tasks.

### 3.2 Modification 1: Using Hindsight Experience Replay (HER)

Numerous technical innovations have been developed for sparse-reward goal-conditioned RL (see Section 6 for details), and many of these innovations can be applied to REDQ. We want to keep our method simple

---

[3]While REDQ was proposed in 2021, it is still one of the best (most sample-efficient) methods for continuous control tasks (see Ball et al. (2023) for example).

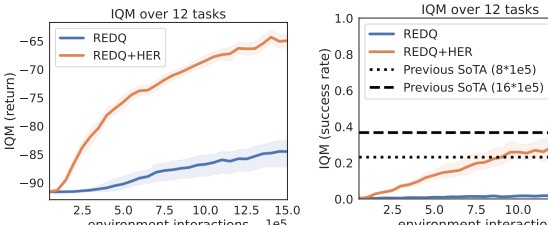

Figure 3: Effect of HER on REDQ's performance. The left figure: the learning curve for a return. The right figure: the curve for task success rate. These figures indicate that the use of HER significantly improves performance. We can see that REDQ with HER (REDQ+HER) exhibits superior returns and success rates to vanilla REDQ.

and flexible to allow its users to introduce complex innovations as needed. Thus, we begin our modification of REDQ by introducing the fundamental component commonly used in previous innovations.

We introduce HER (Andrychowicz et al., 2017) with a future strategy into REDQ to improve its performance. HER with the future strategy is commonly used in previous works for sparse-reward goal-conditioned RL methods (Andrychowicz et al., 2017; Plappert et al., 2018; Zhao & Tresp, 2018; Zhao et al., 2019; Xu et al., 2023). HER replaces a goal $g$ of a past transition with a new goal $g'_t$ to obtain positive rewards (line 5 in Algorithm 1). For selecting the new goal $g'_t$, our HER follows the future strategy. In the future strategy, for each transition $(s_t, a_t, r'_t, s_{t+1}, g) \in \{(s_t, a_t, r'_t, s_{t+1}, g)\}_{t=0}^{T}$, $g$ is replaced with $g'_t$, which is the achieved goal included within a state randomly selected from $\{s_{t+1}, ..., s_T\}$. HER with the future strategy significantly improves REDQ's performance (Fig. 3).

### 3.3 Modification 2: Bounding Target Q-Value (BQ)

REDQ (Section 3.1) employs (i) off-policy learning, (ii) approximation of the value function, and (iii) bootstrapping (i.e. the deadly triad (Sutton & Barto, 2018)). This deadly triad often leads to Q-value estimate divergence and consequently degrades performance (Van Hasselt et al., 2018).

We observe that introducing HER to REDQ induces a divergence in its Q-value estimation. We assess the extent to which Q-value estimates exceed the theoretical upper bound $Q_{\max}$ and lower bound $Q_{\min}$. Here, $Q_{\max}$ is the discounted future return in the best-case scenario, where an agent consistently receives a positive reward, while $Q_{\min}$ is the return in the worst-case scenario with consistent negative rewards. In Robotics (Plappert et al., 2018; de Lazcano et al., 2023) tasks, the positive reward is 0, and the negative reward is -1 [4]. Thus, we estimate $Q_{\max}$ and $Q_{\min}$ as: For any time step $t$, $Q_{\max} = \sum_{t'=t}^{\infty} \gamma^{(t'-t)} \cdot 0 = 0$, and $Q_{\min} = \sum_{t'=t}^{\infty} \gamma^{(t'-t)} \cdot -1 = -1/(1-\gamma)$. The result (Fig. 4) shows that HER induces a divergence in Q-value estimation. We can see that the Q-value estimates of REDQ with HER (REDQ+HER) significantly surpass theoretical bounds compared with those of REDQ.

We bound the target Q-value to mitigate the Q-value estimate divergence. Specifically, we bound the target Q-value using $Q_{\max}$ and $Q_{\min}$ (line 9 in Algorithm 1) [5]:

$$y = r + \gamma \min \left( \max \left( \min_{i \in \mathcal{M}} Q_{\bar{\phi}_i}(s', a', g), Q_{\min} \right), Q_{\max} \right) - \alpha \log \pi_\theta(a'|s', g). \tag{1}$$

Here, $Q_{\max}$ and $Q_{\min}$ are the same as the ones introduced in the preceding paragraph. This bounding effectively suppresses the Q-value estimate divergence (Fig. 4). We will experimentally show that this modification substantially enhances overall performance in the next section.

---

[4]See the second paragraph in Section 2 for a reminder of the reward structure of Robotics tasks.

[5]Note that the idea of bouding target Q-values with worst/best case returns is not new; it has been applied to DDPG (Lillicrap et al., 2015) in the previous works on HER (e.g., Andrychowicz et al. (2017); Zhao & Tresp (2018; 2019); Xu et al. (2023)).

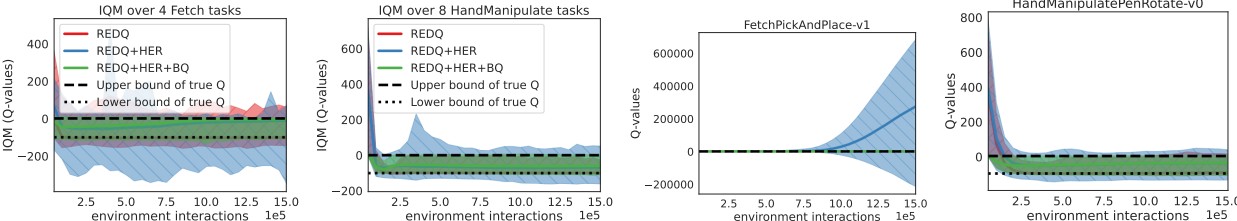

Figure 4: The effect of BQ on Q-value divergence. The solid line represents the average Q-value estimate, while the shaded area represents the range of Q-value estimates. Dashed and dotted lines represent the theoretical upper bound ($Q_{\max}$) and lower bound ($Q_{\min}$) of Q-value, respectively. Two figures on the left-hand side: a summary (IQM) of the scores over 4 Fetch tasks and 8 HandManipulate tasks. Two figures on the right-hand side: examples of scores in individual tasks (FetchPickAndPlace and HandmanipulatePenRotate): From the figures, we can see that (i) Q-value estimates of REDQ with HER (REDQ+HER) significantly exceed the bound range and (ii) estimates of the method using bounded target Q-value (REDQ+HER+BQ) are kept almost within the range. The results for all tasks are shown in Fig. 13 in the appendix.

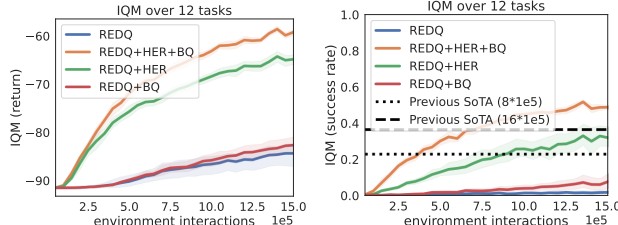

Figure 5: IQM of performance (return and success rate) over 12 Robotics tasks. The left-hand side figure: the IQM of return. The right-hand side figure: the IQM of task success rate. The left-hand side figure shows that REDQ with our modifications (REDQ+HER+BQ) achieves significantly better returns than the others over the 12 tasks. The right-hand figure shows that REDQ+HER+BQ achieves approximately 2× better sample efficiency than previous SoTA methods.

## 4 Experiment

In the previous section, we introduced HER and BQ into REDQ. In this section, we conduct experiments to answer two questions:

**Q1.** Does introducing both HER and BQ enhance REDQ's performance more than introducing HER or BQ individually?

**Q2.** Does REDQ with HER and BQ achieve equal or superior performance compared to previous SoTA methods?

**Experiment for Q1: Our experimental result indicates that introducing both HER and BQ enhances the performance more than introducing HER or BQ individually.** We conduct experiments to evaluate three methods: (i) REDQ+HER+BQ: REDQ using HER and BQ, (ii) REDQ+HER: REDQ using HER alone, and (iii) REDQ+BQ: REDQ using BQ alone. We record the average return over 100 test episodes for each 50000 environment steps, and use it for measuring performance for the methods. The experimental results (the left-hand side figure of Figs. 5) show that introducing both HER and BQ to REDQ (REDQ+HER+BQ) achieves better returns than introducing HER and BQ individually (REDQ+HER and REDQ+BQ), over the 12 Robotics tasks. Results for each task (Fig. 6) show that this synergistic effect of HER and BQ in Fetch tasks tends to be more significant than that in HandManipulate tasks. This likely occurs because BQ more significantly suppresses the Q-value-estimation divergence induced by HER in Fetch tasks compared to HandManipulate tasks (Fig. 13 in the appendix).

**Experiment for Q2: Our experimental result indicates that REDQ with HER and BQ achieves superior performance compared to SoTA RL methods.** We compare REDQ+HER+BQ with previous SoTA methods. Previous SoTA methods denote the best among previous methods. Previous methods are

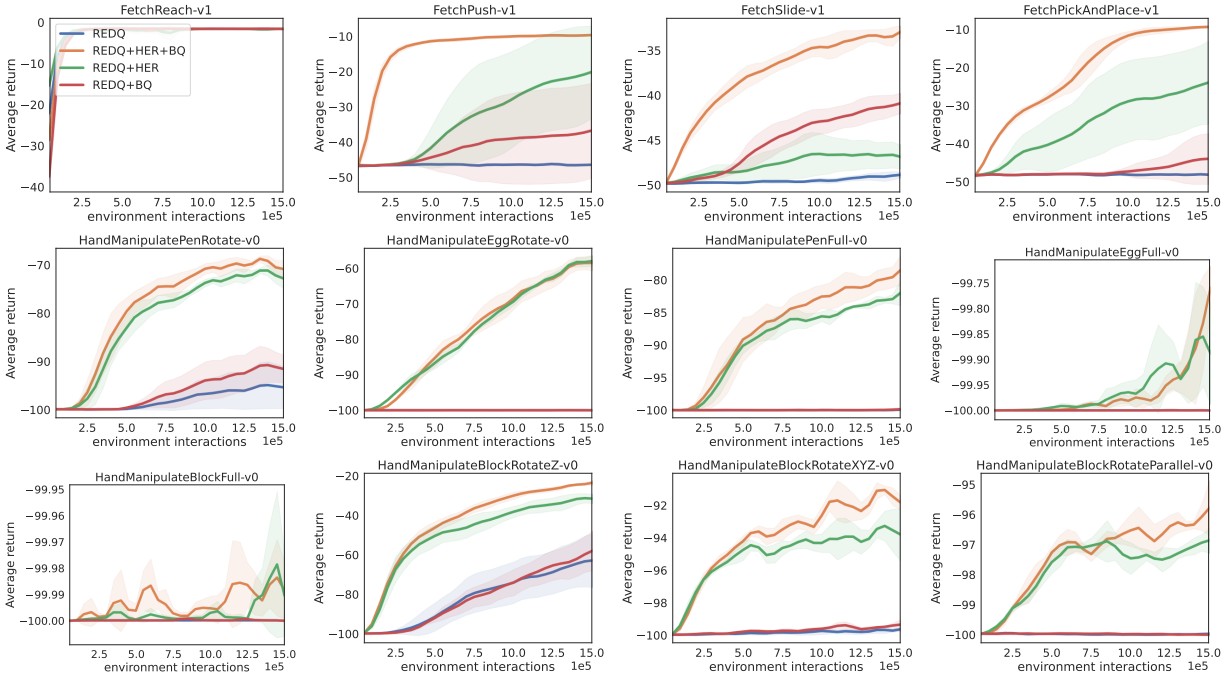

Figure 6: Return improvement in each of 12 Robotics tasks. Figures show that HER alone significantly contributes to the return improvement in HandManipulate tasks, whereas both HER and BQ significantly contribute to the improvement in the Fetch tasks (except for FetchReach).

HER (with DDPG) (Andrychowicz et al., 2017), EBP (Zhao & Tresp, 2018), CHER (Zhao & Tresp, 2019), DTGSH (Dai et al., 2021), and VCP (Xu et al., 2023) [6]. We use the performance score of the best one among them for the performance score of previous SoTA methods, with $8 \cdot 10^5$ and $16 \cdot 10^5$ samples, at each task [7]. As in these previous works, we use a task success rate as a score for measuring the performance of the methods. The experimental results (the right-hand side figure in Figs. 5) show that REDQ+HER+BQ achieves about $2\times$ better sample efficiency than the previous SoTA. REDQ+HER+BQ with $8 \cdot 10^5$ samples performs comparably to the previous SoTA with $16 \cdot 10^5$ samples. In addition, REDQ+HER+BQ with $4 \cdot 10^5$ samples performs comparably to the previous SoTA with $8 \cdot 10^5$ samples. Looking at scores in each task (Fig. 7), REDQ+HER+BQ makes particularly significant improvements against previous SoTA in, e.g., FetchSlide and HandManipulateBlockRotateZ tasks. On the other hand, the success rate of REDQ+HER+BQ is consistently close to 0, similar to the previous SoTA, in very difficult tasks such as HandManipulateEggFull and HandManipulateBlockFull.

## 5 Simplifying Our Method (REDQ+HER+BQ)

In the previous section, we demonstrated the efficacy of REDQ+HER+BQ. However, REDQ+HER+BQ is more complicated than REDQ as it uses additional components (HER and BQ). In this section, we attempt to simplify REDQ+HER+BQ, by removing (or replacing) components of REDQ. Specifically, we attempt to remove (i) clipped double Q-learning and an entropy term, (ii) high RR and regularization, and to replace (iii) REDQ (i.e., all components of REDQ) with a simpler RL method.

**(i) Are clipped double Q-learning and an entropy term removable? Yes.** REDQ calculates the target Q-value (Eq. 1) with clipped double Q-learning (CDQ) (Fujimoto et al., 2018) and an entropy term: (i) CDQ $\min_{i \in \mathcal{M}} Q_{\bar{\phi}_i}(s', a', g)$, and (ii) the entropy term $\alpha \log \pi_\theta(a'|s', g)$. The effectiveness of these components often depends heavily on tasks (Ball et al., 2023). Thus, we investigate whether they are removable in our

---

[6] All of these methods use DDPG (Lillicrap et al., 2015) with a low RR ($\leq 1$) and no regularization, unlike REDQ.

[7] Scores for all of the previous methods are documented in Appendix A.2.

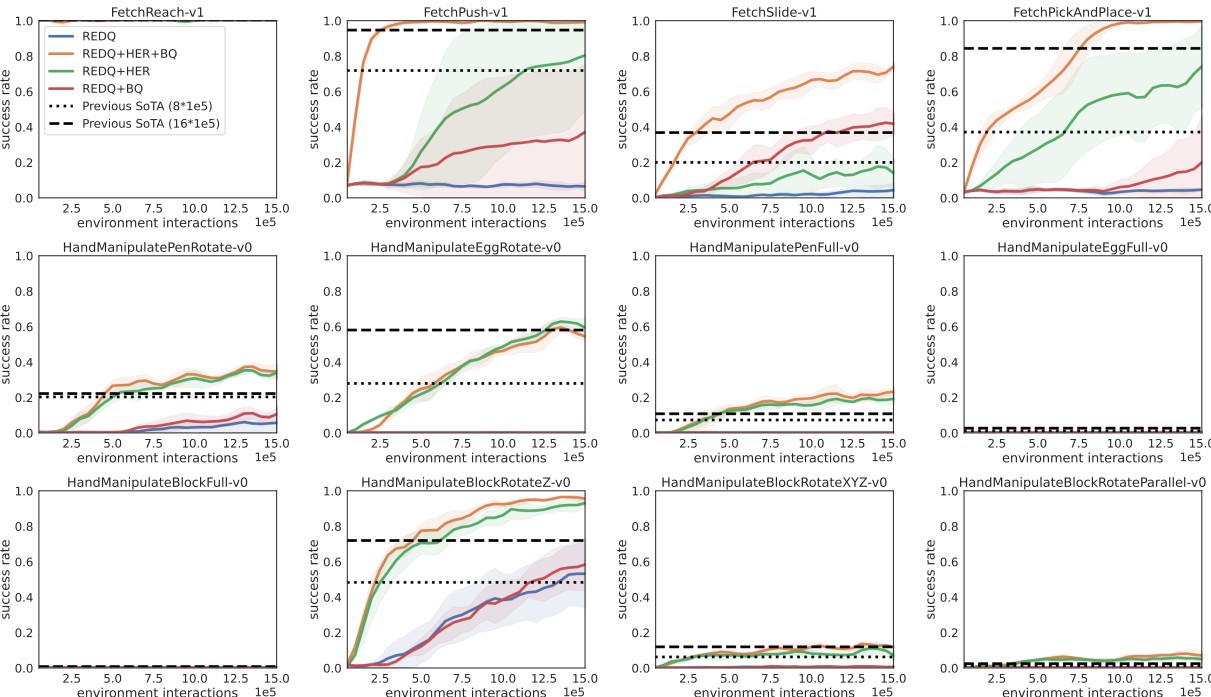

Figure 7: The success rate in 12 Robotics tasks. The figures show that REDQ+HER+BQ exhibits particularly significant improvements compared with previous SoTA methods in the FetchSlide and HandManipulateBlockRotateZ tasks.

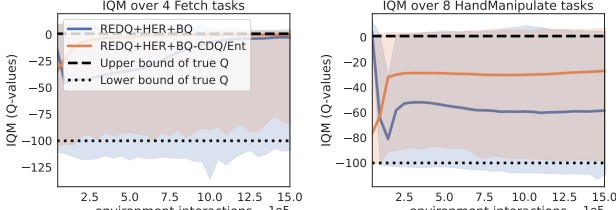

Figure 8: The effect of removing CDQ and the entropy term on Q-value divergence. The figure shows that the method simplified by removing them (REDQ+HER+BQ-CDQ/Ent) can suppress the divergence of the Q-value to a similar extent as the method without the simplification (REDQ+HER+BQ). The results for all tasks are shown in Fig. 17 in the appendix.

task or not. We remove CDQ and the entropy term as:

$$y = r + \gamma \min \left( \max \left( \frac{1}{|\mathcal{M}|} \sum_{i \in \mathcal{M}} Q_{\bar{\phi}_i}(s', a', g), Q_{\min} \right), Q_{\max} \right). \tag{2}$$

Here, the average operator $\frac{1}{|\mathcal{M}|} \sum_{i \in \mathcal{M}}$ is used instead of the minimum operator $\min_{i \in \mathcal{M}}$. The method simplified in this way (REDQ+HER+BQ-CDQ/Ent) can suppress Q-value divergence to a similar extent to the original method (REDQ+HER+BQ) (Fig. 8). In addition, REDQ+HER+BQ-CDQ/Ent can achieve almost the same overall (IQM) performance as REDQ+HER+BQ (the left-hand side figure in Figs. 9). Furthermore, REDQ+HER+BQ-CDQ/Ent achieves $\sim 8\times$ better sample efficiency than the previous SoTA in the FetchPickAndPlace task (the right-hand side figure in Figs. 9). Given these results, we conclude that CDQ and the entropy term are removable in our tasks.

**(ii) Are a high RR and regularization removable? No.** So far, we have considered several design choices. Even after this consideration, are the core components of REDQ (i.e., a high RR and regularization) (Section 3.1) still necessary for our method? To answer this question, we evaluate two variants of

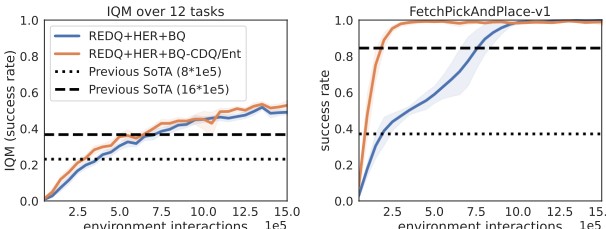

Figure 9: The effect of removing CDQ and the entropy term on performance. The left-hand side figure: the IQM scores over 12 Robotics tasks. The right-hand side figure: the average return and success rate in FetchPickAndPlace. The left-hand side figure shows that the method not using CDQ and entropy term (REDQ+HER+BQ-CDQ/Ent) achieves an overall performance comparable to that of the original method (REDQ+HER+BQ). The right-hand side figure shows that REDQ+HER+BQ-CDQ/Ent achieves $\sim 8\times$ better sample efficiency than the previous SoTA in the FetchPickAndPlace task. The results for all tasks are shown in Figs. 18 and 19 in the appendix.

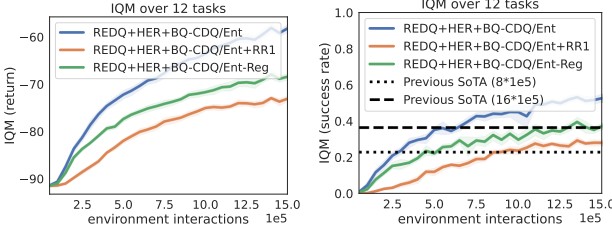

Figure 10: The effect of removing a high RR and regularization on performance. The figure (IQM scores over 12 tasks) shows that both a high RR and regularization are necessary. The results for all tasks are shown in Figs. 20 and 21 in the appendix.

REDQ+HER+BQ-CDQ/Ent that do not use a high RR and regularization:
1. REDQ+HER+BQ-CDQ/Ent+RR1: The method without a high RR. It uses a low RR of 1.
2. REDQ+HER+BQ-CDQ/Ent-Reg: The method without regularization. It uses a small ensemble (i.e., two Q-functions) and no layer normalization.
The evaluation results (Fig. 10) show that both a high RR and regularization are still necessary for our method. We can see that REDQ+HER+BQ-CDQ/Ent achieves better sample efficiency than REDQ+HER+BQ-CDQ/Ent+RR1 and REDQ+HER+BQ-CDQ/Ent-Reg.

**(iii) Can REDQ be replaced with a simpler method (Reset (Nikishin et al., 2022))? No.** There are RL methods other than REDQ that have a high RR and regularization (Section 6). Can we use these other methods, especially a simple one, instead of REDQ for our base RL method? To answer this, we compare REDQ-based methods (e.g., REDQ+HER+BQ or REDQ+HER) with methods based on Reset (Nikishin et al., 2022). Reset does regularization simply by periodically initializing the parameters of the agent's components (policy and Q-functions). Despite its simplicity, it performs equally to or better than REDQ in some dense-reward continuous-control tasks (D'Oro et al., 2023). We use four Reset-based methods for our comparison:
1. Reset([the number of resets]): Reset (Nikishin et al., 2022) itself. "[number of resets]" means the total number of resets during training. In our experiments, we use Reset(1), Reset(4), and Reset(9). In addition, we use an RR of 20 as with our REDQ.
2. Reset([the number of resets])+HER: Reset with HER.
3. Reset([the number of resets])+BQ: Reset with BQ.
4. Reset([the number of resets])+HER+BQ: Reset with HER and BQ.
The algorithmic description of these Reset-based methods is summarized in Algorithm 2 in the appendix. The comparison results (Fig. 11) show that REDQ is more suitable for our base RL method than Reset *in our setting*. We can see that REDQ+HER+BQ performs better than other Reset-based methods.
**Complementary analysis: Does introducing HER and BQ into Reset improve or at least keep its performance? Yes.** From Fig. 11, we can see the following trends: (i) Reset(1, 4, 9)+HER achieves better sample efficiency than Reset(1, 4, 9). (ii) Reset(1, 4, 9)+HER+BQ achieves the same or better

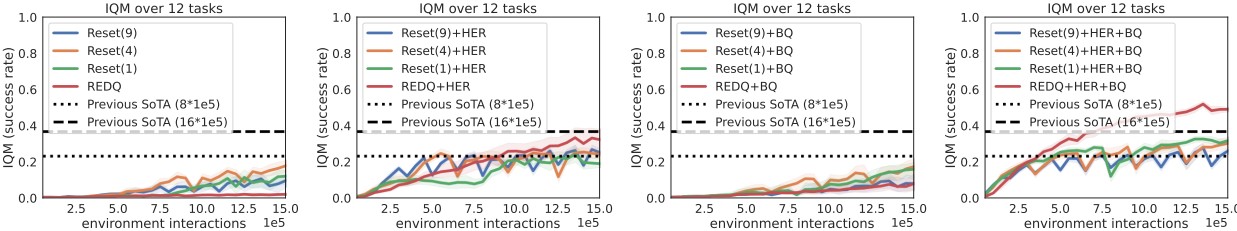

Figure 11: The effect of replacing REDQ with Reset (Nikishin et al., 2022) on performance (success rate). The figure shows that REDQ is more suitable for our base RL method than Reset. We can see that REDQ+HER+BQ achieves better performance than other Reset-based methods. The results for all tasks are shown in Figs. 22, 23, 24, and 25 in the appendix.

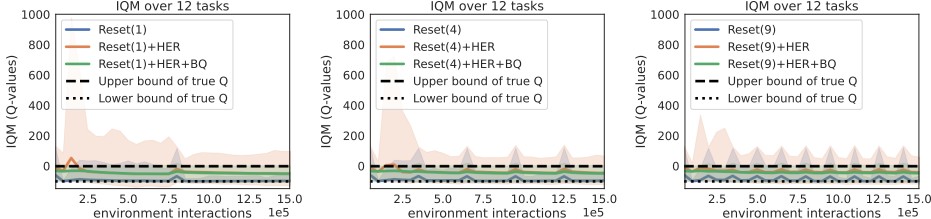

Figure 12: The effect of replacing REDQ with Reset on Q-value divergence. The figures show that HER causes Q-estimation divergence more significantly for the Reset method with a smaller reset number. We can see that Q-estimates of Reset(1)+HER exceed the bound range more significantly than Reset(4, 9)+HER. The results for all tasks are shown in Figs. 26, 27, and 28 in the appendix.

sample efficiency than Reset(1, 4, 9)+HER. Especially, Reset(1)+HER+BQ achieves significantly better sample efficiency than Reset(1)+HER.

## 6 Related Works

**RL methods with a high RR and regularization.** We applied RL methods with a high RR and regularization to sparse-reward tasks (Sections 3 and 4). Most previous works on RL methods with a high RR and regularization have focused primarily on dense-reward tasks (Janner et al., 2019; Kumar et al., 2021; Chen et al., 2021; Hiraoka et al., 2022; Smith et al., 2022; Nikishin et al., 2022; Li et al., 2023a; D'Oro et al., 2023; Smith et al., 2023b; Sokar et al., 2023; Schwarzer et al., 2023; Lee et al., 2023). Some works (Vecerik et al., 2017; Sharma et al., 2023; Ball et al., 2023; Nakamoto et al., 2023; Li et al., 2023b) have considered sparse-reward tasks, but they assume situations where prior data (e.g., expert demonstrations) are available. On the other hand, we assume sparse-reward tasks where such prior data are unavailable. Our work is orthogonal to the above previous works, and some of our modifications may be useful in them. For example, we bound the target Q-value to deal with the instability of Q-value estimation (Section 3.3), and a similar instability problem also appears in the above works (see Fig. 2 in Ball et al. (2023) for example).

**Sparse-reward goal-conditioned RL.** There are previous works on sparse-reward goal-conditioned RL. These works have used DDPG (Lillicrap et al., 2015) (or soft actor-critic (Haarnoja et al., 2018)) with HER (Andrychowicz et al., 2017) as a base RL method and improved its sampling prioritization scheme (Zhao & Tresp, 2018; Zhao et al., 2019; Zhao & Tresp, 2019; Dai et al., 2021; Xu et al., 2023), relabeling scheme (Yang et al., 2021), and new-goal-selection strategies (Fang et al., 2019; Pitis et al., 2020; Ren et al., 2019; Chane-Sane et al., 2021; Luo et al., 2022). In these works, the base RL method uses low RR (≤ 1) and no regularization, which is contrary to our base RL method (Section 3.1). We showed that the RL method with a high RR and regularization can achieve a better sample efficiency than methods with a low RR and no regularization (the right-hand side figure in Figs. 5 in Section 4). **Other related works for sparse-reward tasks:** While we focused on the HER approach in our paper, there are various other approaches for dealing with sparse-reward tasks, e.g., (Pertsch et al., 2020; Singh et al., 2020; Nam et al., 2022; Siegel et al., 2020).

**Bounding Q-value (BQ).** We used BQ for sparse-reward goal-conditioned RL (Section 3.3). Most previous works on sparse-reward goal-conditioned RL have applied BQ together with HER to DDPG (e.g., Andrychowicz et al. (2017); Zhao & Tresp (2018; 2019); Xu et al. (2023)). However, these works have left three points about BQ unclear. First, they did not conduct ablation studies to quantify BQ's contribution to performance enhancement. Second, they did not explain the rationale behind using BQ. Third, they did not evaluate the effectiveness of BQ, especially when used with HER, for RL methods other than DDPG. To clear up these points, we conducted several experiments (Sections 3, 4, and 5). In the context of online RL other than sparse-reward goal-conditioned RL, other previous works have also proposed the bounding of Q-values (Blundell et al., 2016; S.He et al., 2017; Oh et al., 2018; Lin et al., 2018; Tang, 2020; Fujita et al., 2020; Hoppe & Toussaint, 2020; Zhao & Xu, 2023; Fujimoto et al., 2023). These previous works have verified a positive effect of the bounding on RL methods with a low RR and no regularization in dense-reward tasks. Our work has verified a positive effect of the bounding on the RL method with a high RR and regularization in sparse-reward tasks.

**HER bias:** We used BQ to mitigate the Q-functions instability induced by HER (Section 3.3). Previous works have demonstrated that HER introduces biases in transition-data distribution (Lanka & Wu, 2018), which could cause a value-estimation bias (Yang et al., 2021; Blier & Ollivier, 2021; Schramm et al., 2023). Some readers may wonder if the value-estimation bias also occurs in our tasks. Overall, we did not observe a clear appearance of the value-estimation bias induced by HER in our tasks (see Appendix C for a detailed report).

## 7 Conclusion, Limitations, and Future Work

**Conclusion.** We applied a reinforcement learning (RL) method (Randomized Ensemble Double Q-learning; REDQ) with a high replay ratio (RR) and regularization to sparse-reward goal-conditioned tasks. We introduced hindsight experience replay (HER) and bounding target Q-value (BQ) to REDQ and showed that REDQ with them achieves about $2\times$ better sample efficiency than previous state-of-the-art (SoTA) methods in 12 Robotics tasks. We also showed that REDQ with HER and BQ can be simplified by removing clipped double Q-learning (CDQ) and entropy terms. The simplified REDQ with our modifications achieved $\sim 8\times$ better sample efficiency than the SoTA methods in the 4 Fetch tasks of Robotics. We hope that these findings will push the boundaries of the application of RL methods with a high RR and regularization from dense-reward tasks to sparse-reward tasks.

**Limitations and future work.** Our work leaves limitations and future work:

1. Our RL method did not significantly improve the sampling efficiency in extremely hard tasks (e.g., HandManipulateBlockFull). Improving the efficiency in these tasks is an interesting future work.
2. Our experiments are conducted in simulated environments, not real ones. Our primary interest lies more in investigating decision choices for RL methods rather than in demonstration in real environments. Nevertheless, demonstration in real environments would be one of the natural future steps for our work.
3. We focused on the empirical assessment of the effectiveness of our method. Assessment from a theoretical perspective would be an interesting direction for future work.

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

# A Detailed Experimental Results

## A.1 Modification 2: Bounding Target Q-Value

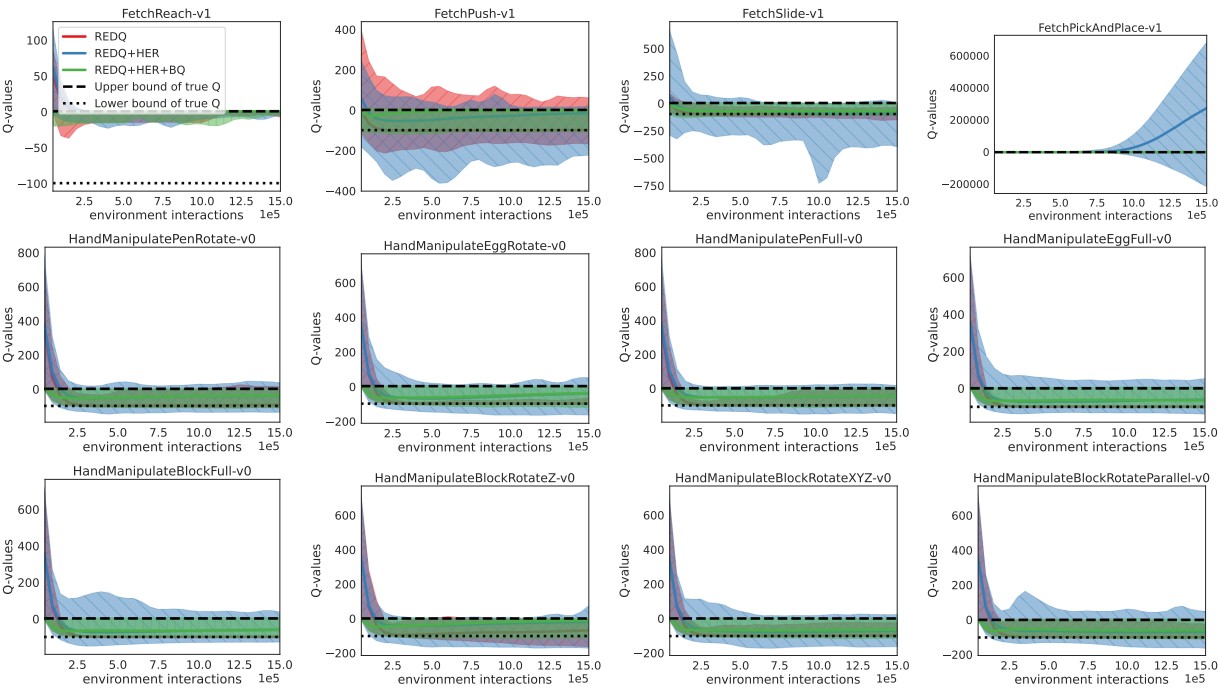

Figure 13: The effect of BQ on Q-value divergence.

## A.2    Scores for Previous Methods

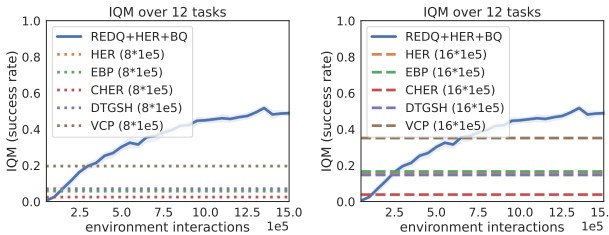

Figure 14: IQM of performance (success rate) over 12 Robotics tasks.

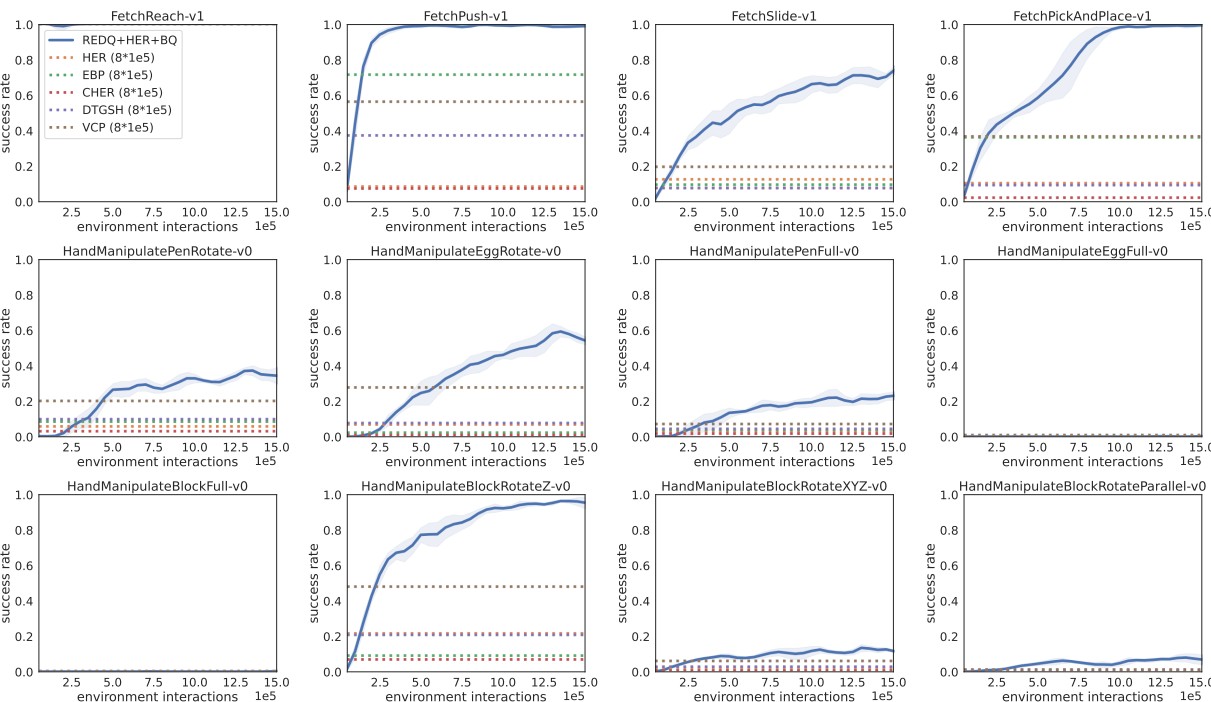

Figure 15: The success rate in 12 Robotics tasks ($8 \cdot 10^5$ samples).

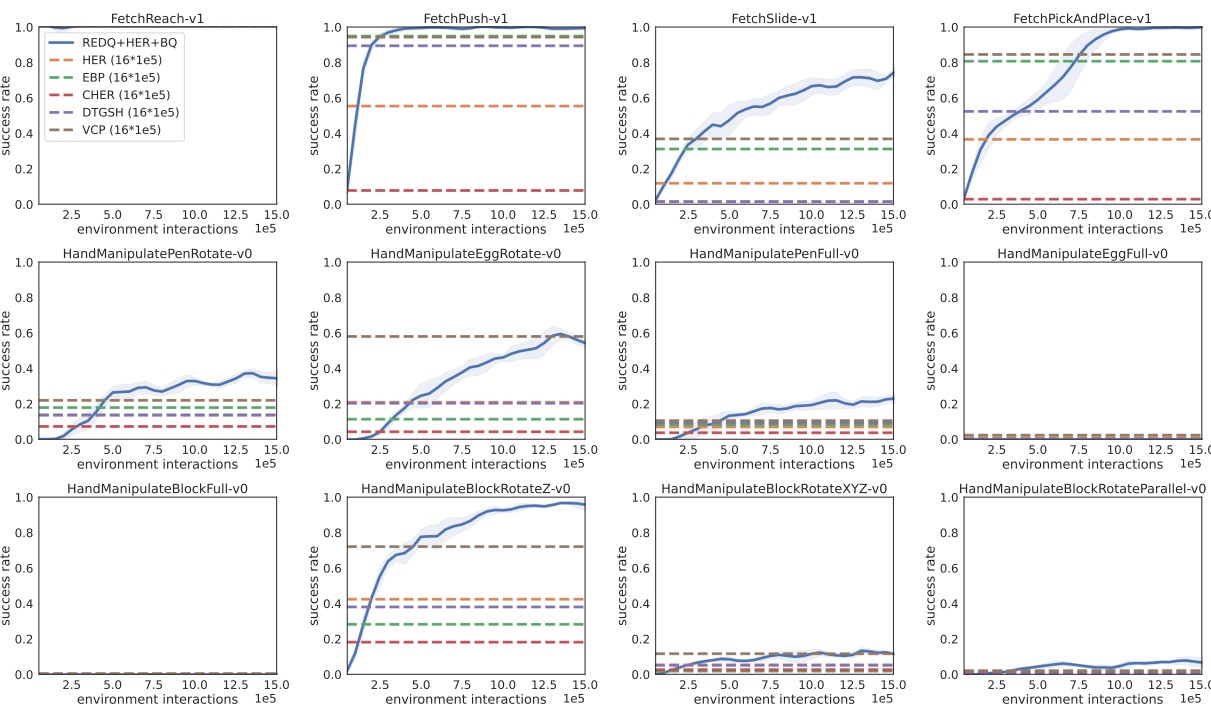

Figure 16: The success rate in 12 Robotics tasks ($16 \cdot 10^5$ samples).

## A.3  Simplifying Our Method (REDQ+HER+BQ)

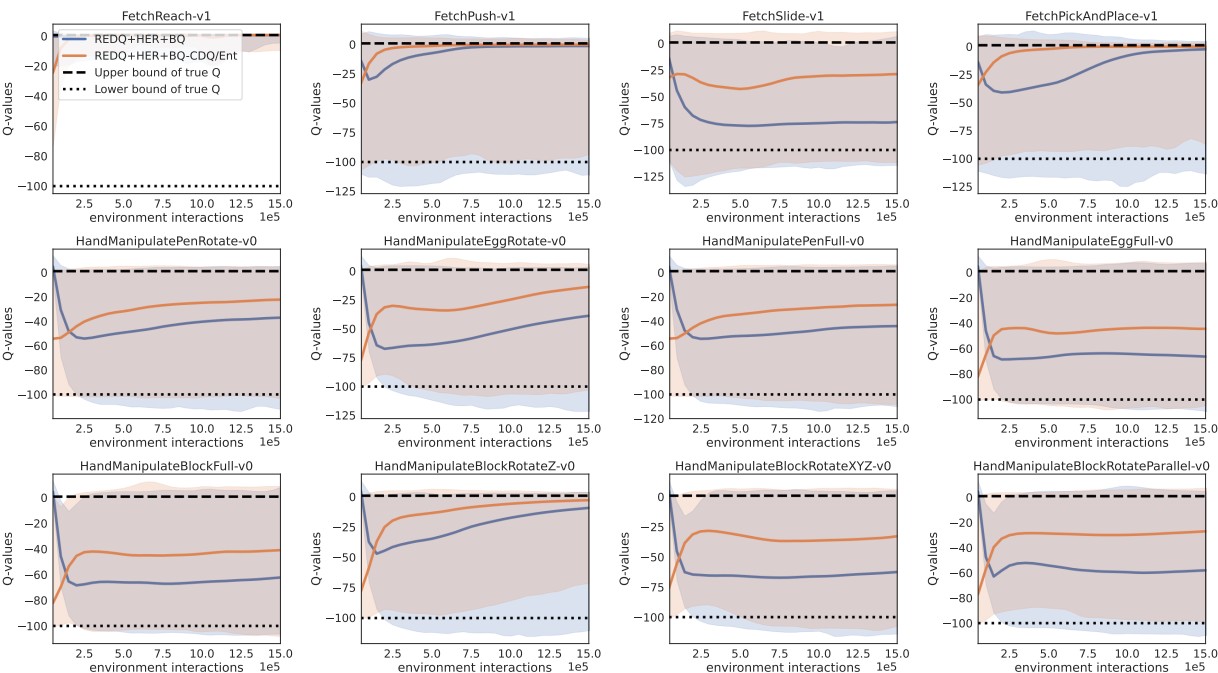

Figure 17: The effect of removing CDQ and entropy term on Q-value divergence.

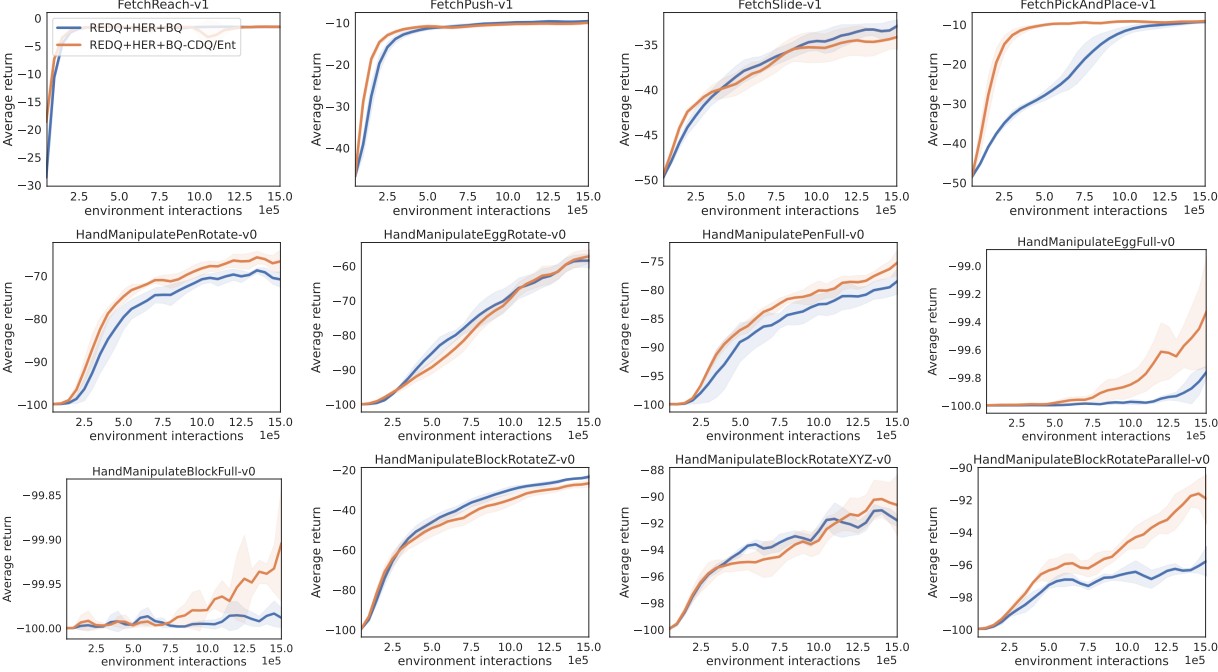

Figure 18: The effect of removing CDQ and the entropy term on performance (return).

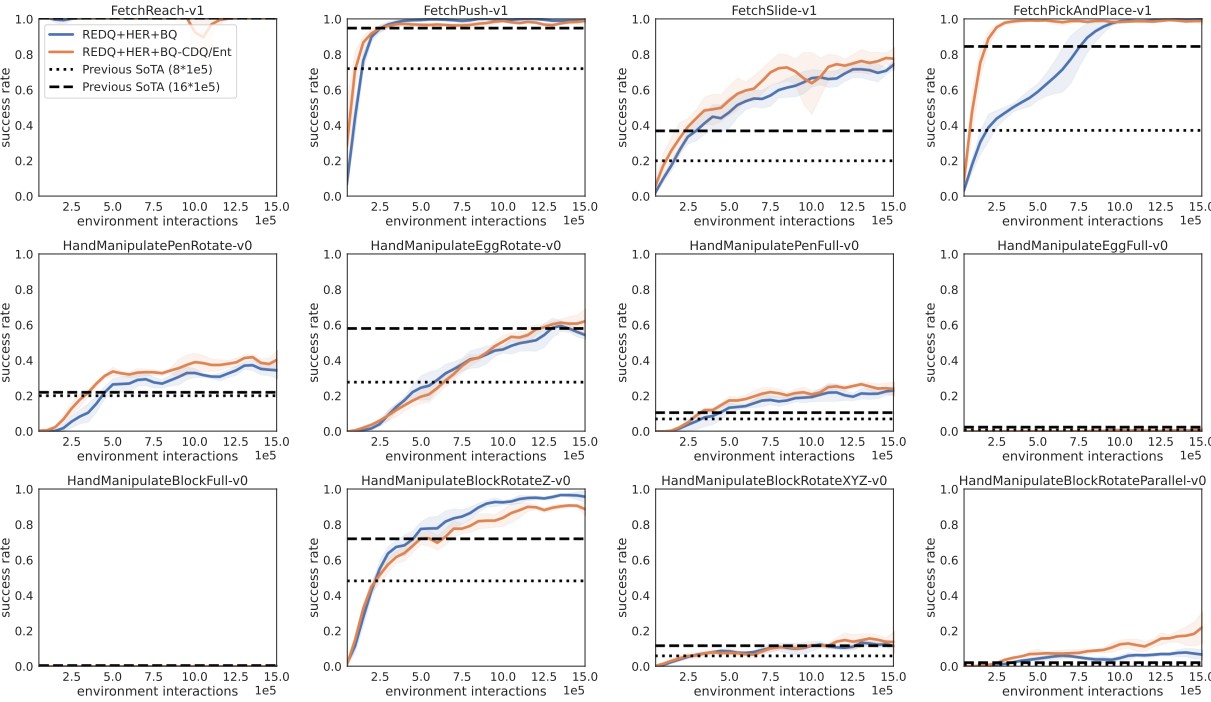

Figure 19: The effect of removing CDQ and the entropy term on performance (success rate).

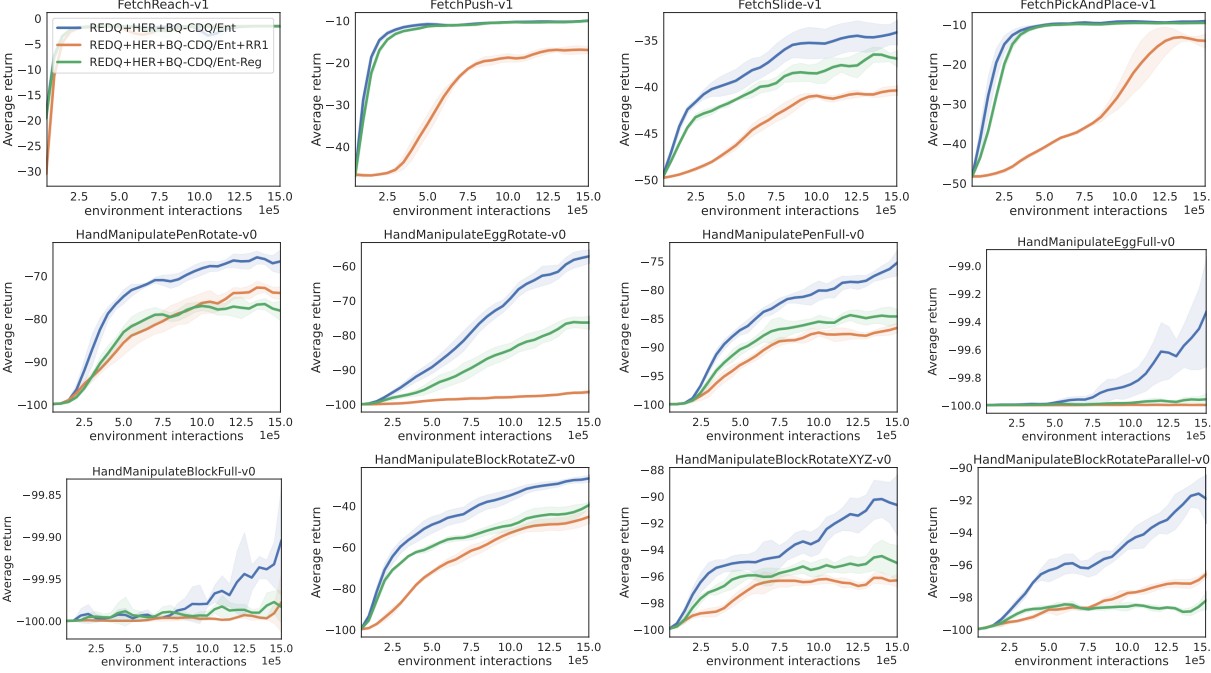

Figure 20: The effect of removing a high RR and regularization on performance (return).

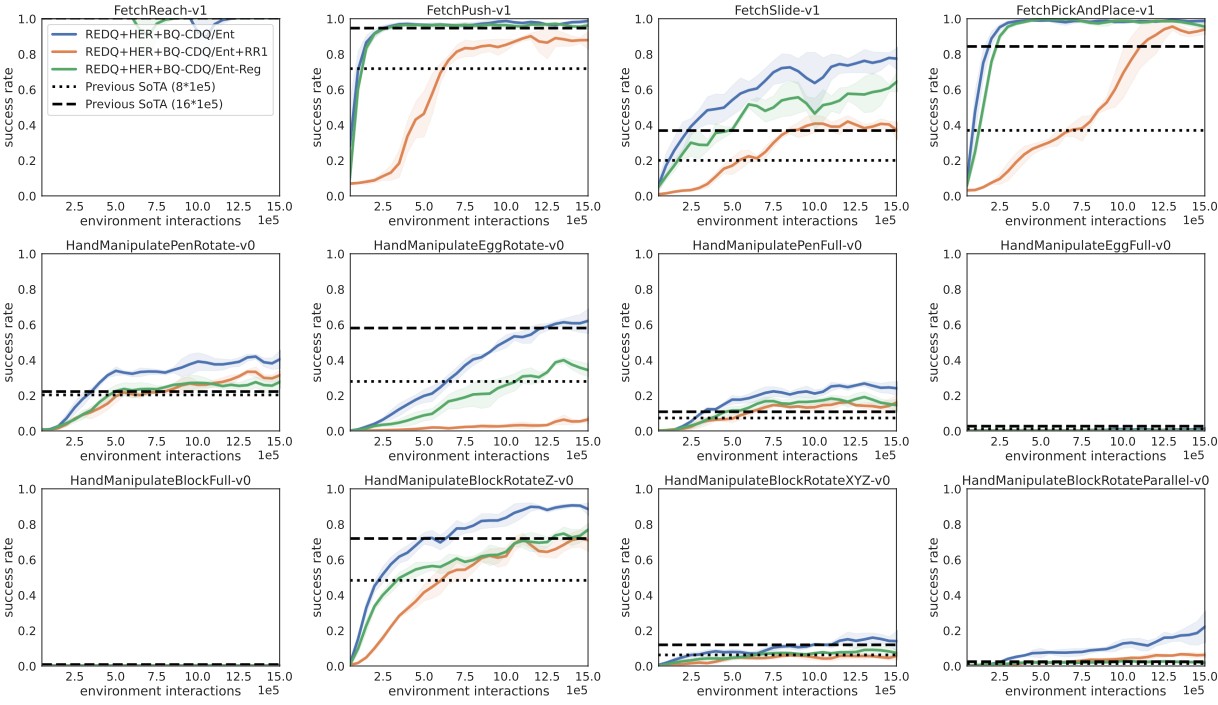

Figure 21: The effect of removing a high RR and regularization on performance (success rate).

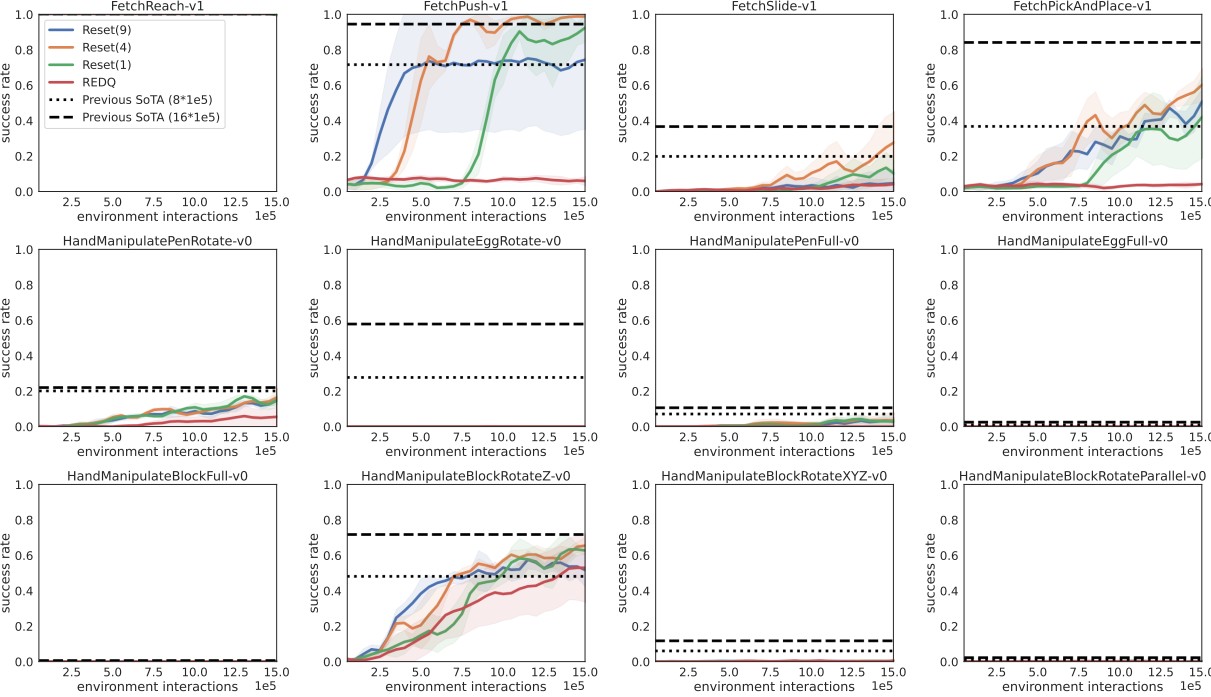

Figure 22: The effect of replacing REDQ with Reset on performance (success rate).

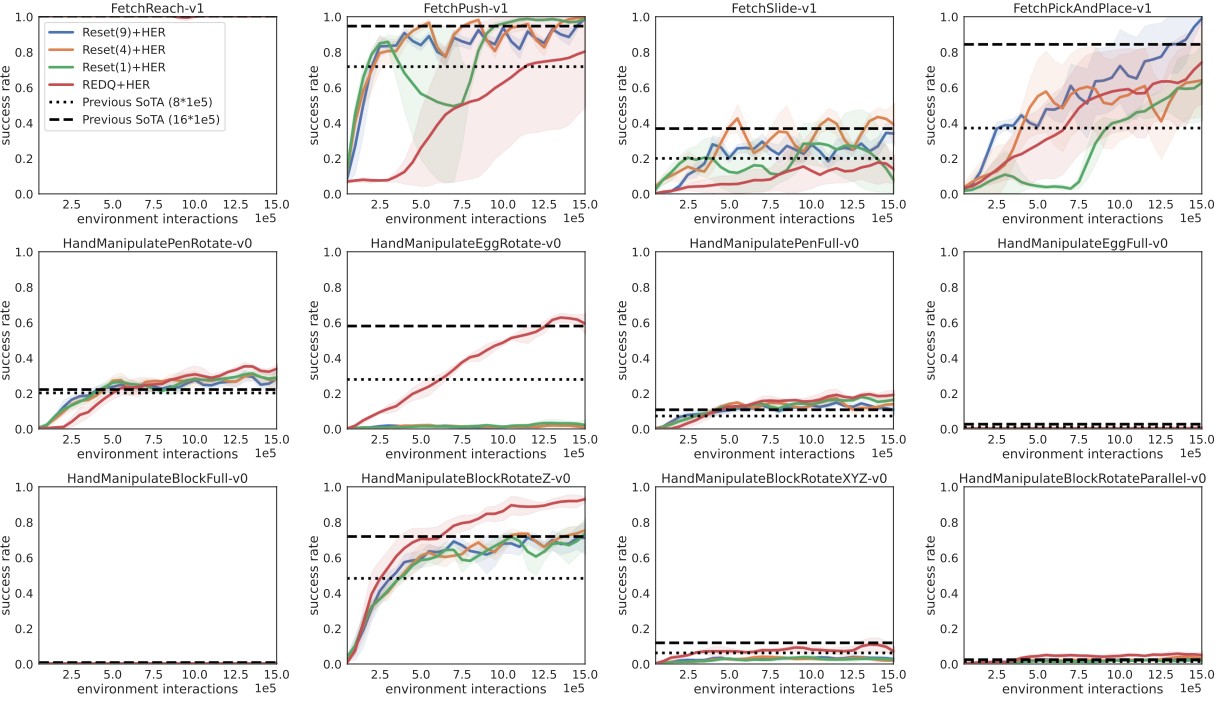

Figure 23: The effect of replacing REDQ+HER with Reset+HER on performance (success rate).

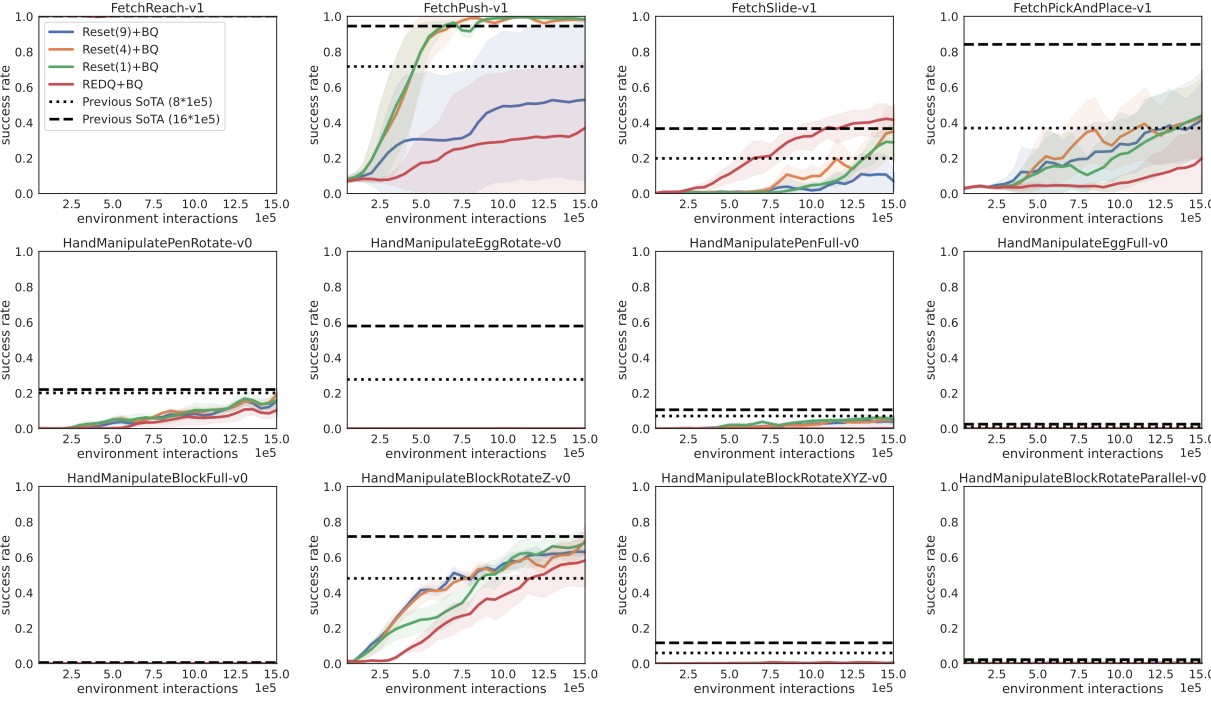

Figure 24: The effect of replacing REDQ+BQ with Reset+BQ on performance (success rate).

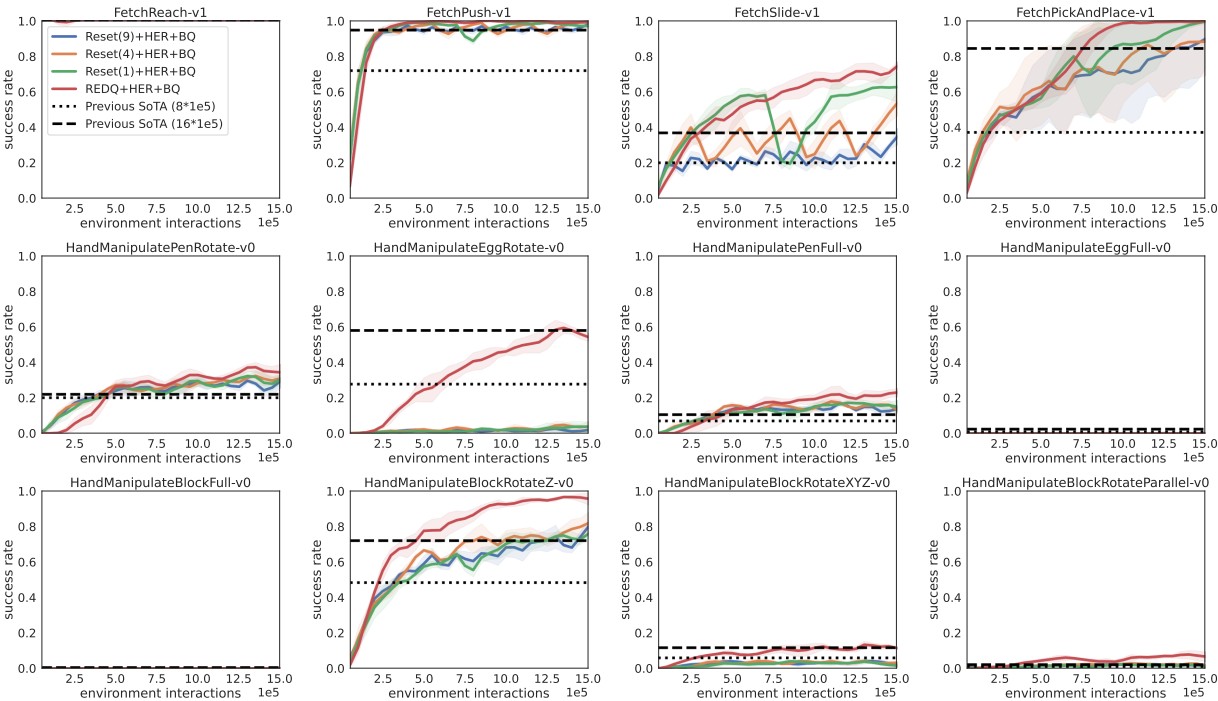

Figure 25: The effect of replacing REDQ+HER with Reset+HER on performance (success rate).

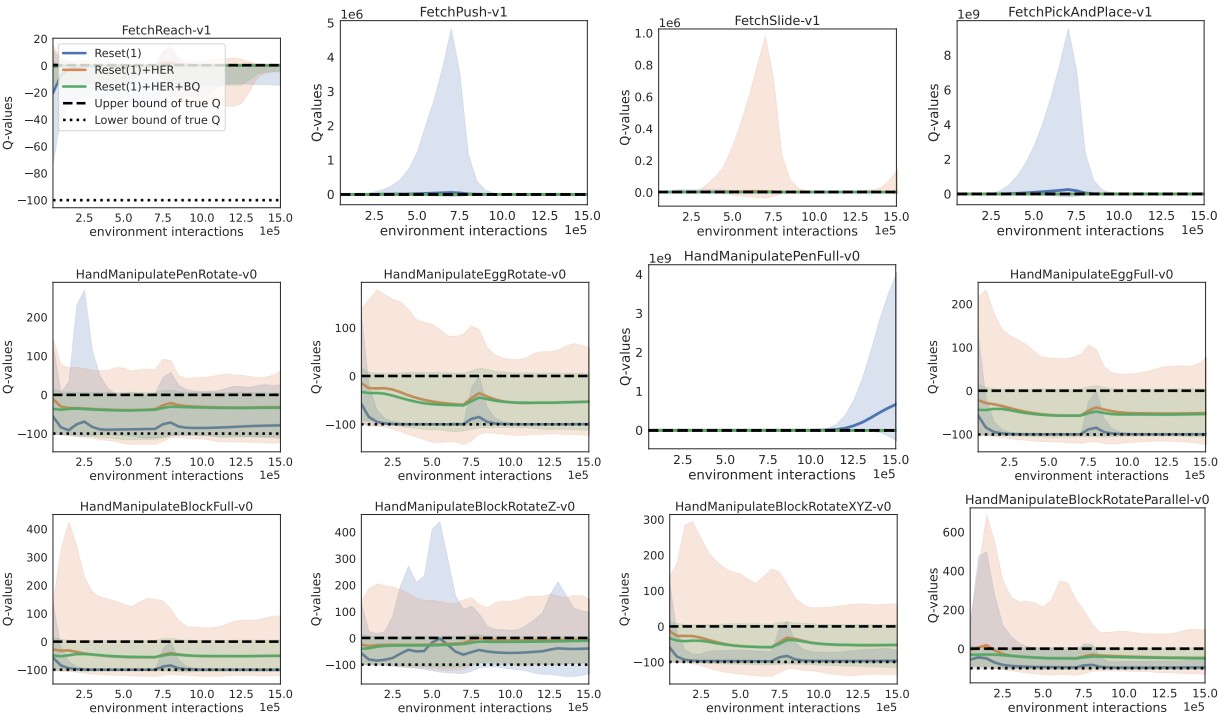

Figure 26: The effect of replacing REDQ with Reset(1) on Q-value divergence.

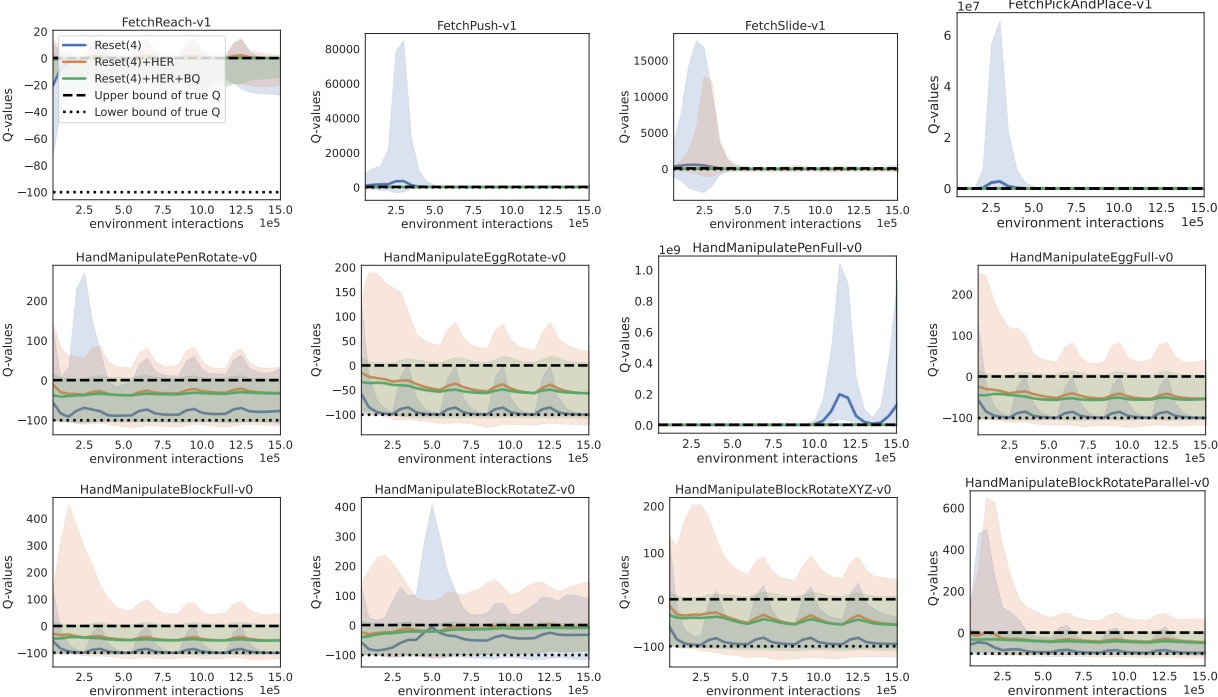

Figure 27: The effect of replacing REDQ with Reset(4) on Q-value divergence.

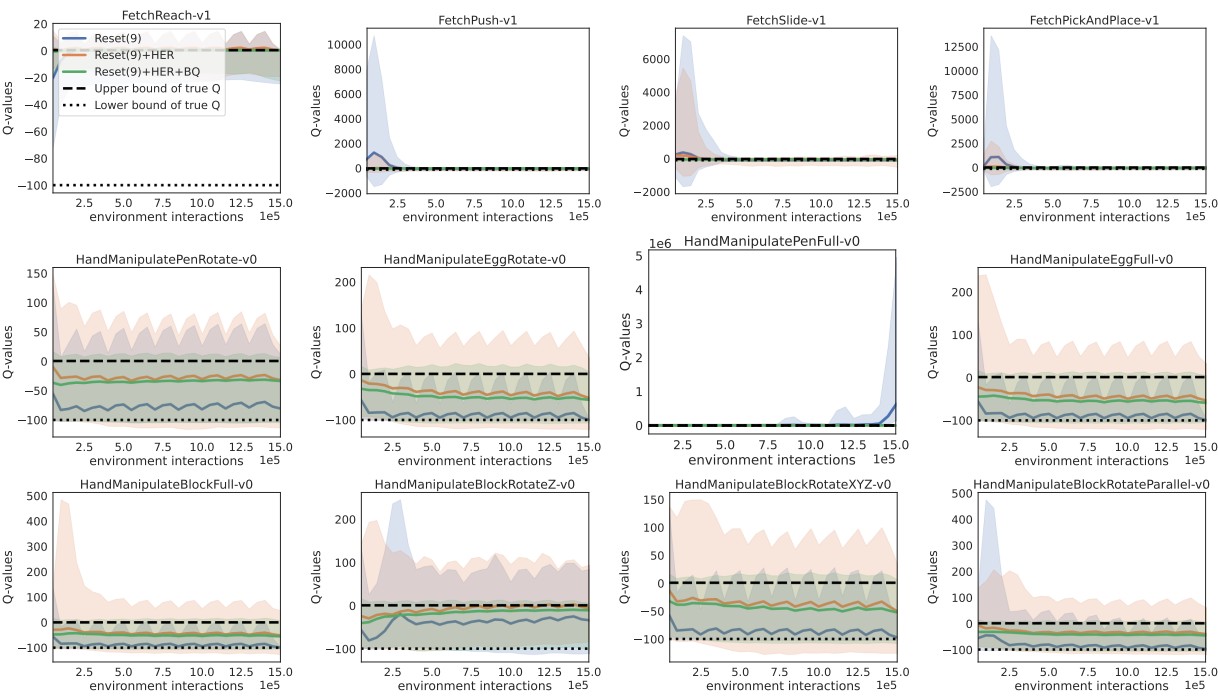

Figure 28: The effect of replacing REDQ with Reset(9) on Q-value divergence.

# B  Other Ways to Bound Q-Values

## B.1  Bounding Q-functions with Auxiliary Losses

Some previous works bound Q-functions instead of target Q-values (Blundell et al., 2016; Oh et al., 2018; Lin et al., 2018; Tang, 2020; S.He et al., 2017). These works use auxiliary losses for Q-function learning to bound the functions. Similar to these works, we consider bounding Q-functions with the auxiliary losses.

We refer to the variant of REDQ+HER+BQ that bounds Q-functions with auxiliary losses as REDQ+HER+BQ(Aux). Instead of the original Q-function learning loss (line 11 in Algorithm 1), REDQ+HER+BQ(Aux) uses the loss augmented with the auxiliary losses:

$$\frac{1}{|B|} \sum_{(s,a,r,s',g) \in \mathcal{B}} (Q_{\phi_i}(s,a,g) - y)^2 + \underbrace{\lambda_1 \max(Q_{\phi_i}(s,a,g) - Q_{\max}, 0)^2 + \lambda_2 \max(Q_{\min} - Q_{\phi_i}(s,a,g), 0)^2}_{\text{Auxiliary losses}}.$$

Here, $\lambda_1$ and $\lambda_2$ are scalar hyperparameters to balance losses. $\max(Q_{\phi_i}(s,a,g) - Q_{\max}, 0)^2$ and $\max(Q_{\min} - Q_{\phi_i}(s,a,g), 0)^2$ are auxiliary losses for regularizing upper bound and lower bound of Q-function outputs, respectively. Note that REDQ+HER+BQ(Aux) does not bound target Q-values.

We evaluated REDQ+HER+BQ(Aux) with two hyperparameter values:
REDQ+HER+BQ(Aux)0.5: REDQ+HER+BQ(Aux) with $\lambda_1 = \lambda_2 = 0.5$.
REDQ+HER+BQ(Aux)0.05: REDQ+HER+BQ(Aux) with $\lambda_1 = \lambda_2 = 0.05$.

The evaluation results are shown in Figs. 29 and 30. Fig. 29 shows that the Q-value estimation of REDQ+HER+BQ(Aux) is converged to the values exceeding the range of specified bounds. Besides, Fig. 30 shows that the performance of REDQ+HER+BQ(Aux) is much lower than those of REDQ+HER+BQ. These results indicate that bounding Q-functions is not a better choice than bounding target Q-value.

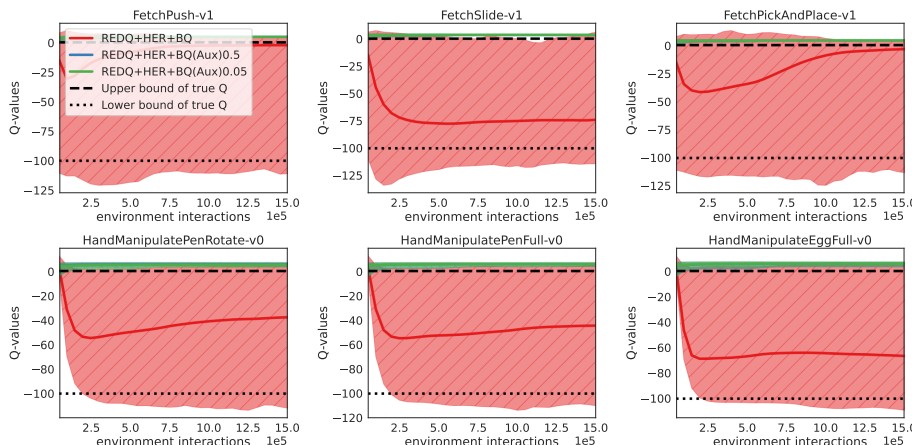

Figure 29: The effect of bounding Q-functions with auxiliary losses (Q-value divergence).

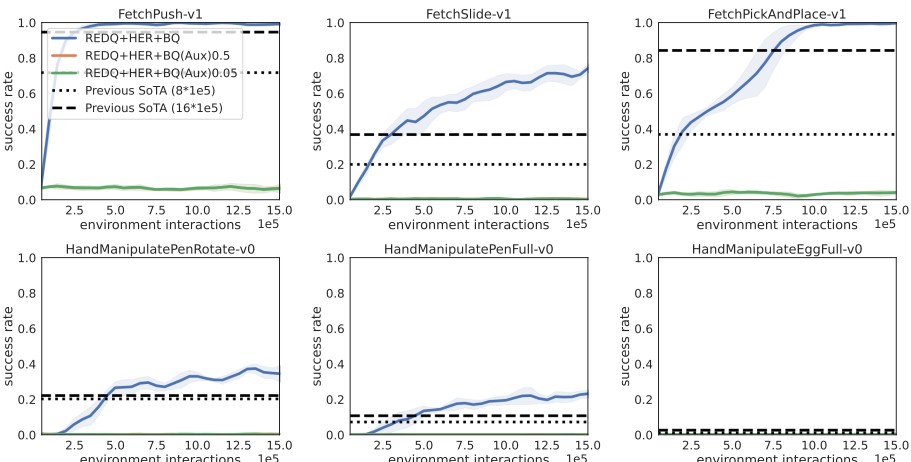

Figure 30: The effect of bounding Q-functions with auxiliary losses (success rate).

## B.2 Bounding Target Q-Values with Empirical Returns

Some previous works use the empirical returns obtained in episodes for the lower bound of target Q-values (Fujita et al., 2020; Zhao & Xu, 2023; Fujimoto et al., 2023). In this section, similar to these works, we consider using empirical returns for the lower bound of target Q-values.

We refer to the variant of REDQ+HER+BQ that uses empirical returns for the lower bound of target Q-values as REDQ+HER+BQ(Ret). REDQ+HER+BQ(Ret) uses the (discounted) empirical return $\sum_{t'=t+1}^{T} \gamma^{(t+1-t')} r_{t'}$ for the lower bound of target Q-values when using training sample $(s_t, a_t, r_t, s_{t+1}, g_t)$ [8]:

$$y = r_t + \gamma \min \left( \max \left( \min_{i \in \mathcal{M}} Q_{\bar{\phi}_i}(s_{t+1}, a_{t+1}, g_t), \sum_{t'=t+1}^{T} \gamma^{(t+1-t')} r_{t'} \right), 0 \right) - \alpha \log \pi_\theta(a_{t+1}|s_{t+1}, g_t),$$
$$a_{t+1} \sim \pi_\theta(\cdot|s_{t+1}, g_t).$$

Besides, for REDQ+HER+BQ(Ret), we convert our task to an episodic task. The bounding Q-value with empirical return requires tasks to be episodic, but the Robotics (Plappert et al., 2018; de Lazcano et al., 2023) tasks we focus on are not episodic. Specifically, in the Robotics tasks, although environments are reset at a terminal timestep $T$, no terminal state is defined, and Q-value is estimated for an infinite planning horizon [9]. We convert the Robotics tasks into an episodic task by including the timestep $t$ in the state and defining the states with $t = T$ as terminal states.

We evaluate REDQ+HER+BQ(Ret) and its result is shown in Figs. 31 and 32. Fig. 31 shows that REDQ+HER+BQ(Ret) consistently reduces Q-value divergence more significantly than the methods (REDQ+HER+BQ) that do not use empirical returns for the lower bound. However, Fig. 32 shows that REDQ+HER+BQ(Ret) does not always achieve higher performance than REDQ+HER+BQ. These results imply that using a stricter lower bound may not necessarily improve performance.

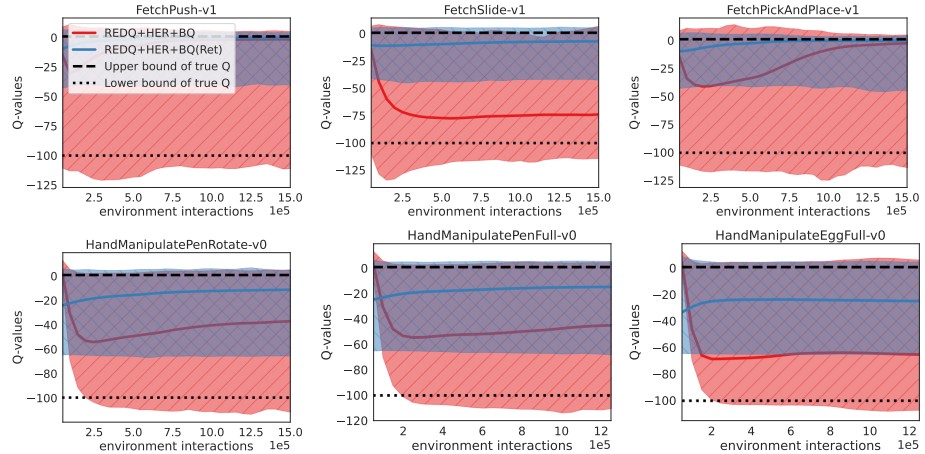

Figure 31: The effect of using empirical returns for lower bound (Q-value divergence).

---

[8]We use 0 for the upper bound as with REDQ+HER+BQ.
[9]See "Episode End" at, e.g., https://robotics.farama.org/envs/fetch/pick_and_place/

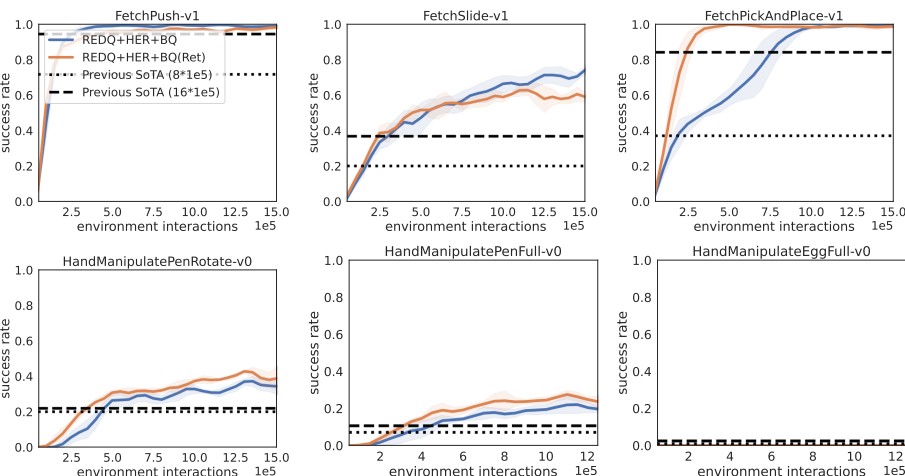

Figure 32: The effect of using empirical returns for lower bound (success rate).

## C  Does HER Induce Value-Estimation Bias?

Previous works (e.g., Schramm et al. (2023)) have shown that HER could induce value-estimation bias. We investigate whether such bias is observed in our task.

For this, we evaluate REDQ and REDQ+HER [10] with the normalized estimation bias (and its standard deviation) (Chen et al., 2021). The bias represents how significantly the Q-value estimate differs from the true one. Formally, it is defined as $|Q^\pi(s,a,g) - \hat{Q}(s,a,g)|/\mathbb{E}_{\bar{s},\bar{a}\sim\pi}[Q^\pi(\bar{s},\bar{a},g)]$, where $Q^\pi(s,a,g)$ is the true Q-value under the current policy $\pi$ and $\hat{Q}(s,a,g)$ is its estimate. In our evaluation, $Q^\pi(s,a,g)$ was approximated by the discounted Monte Carlo return obtained with $\pi$ in test episodes.

The experimental results (Fig. 33) show that, overall, there is no clear appearance of the value-estimation bias induced by HER in our tasks. We can see that the bias and its standard deviation for REDQ+HER significantly overlap with those for REDQ [11]. Our results are consistent with insights presented in previous works. Our tasks (i.e., Robotics (Plappert et al., 2018) tasks) are deterministic tasks where state transition is deterministic. It is known that, in deterministic tasks, the HER bias does not manifest significantly (Plappert et al., 2018; Blier & Ollivier, 2021).

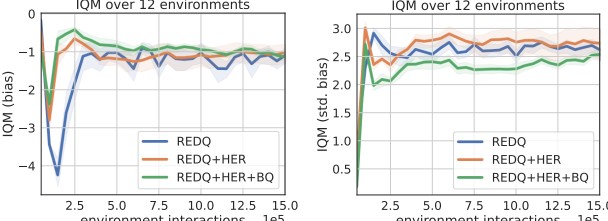

Figure 33: IQM of estimation bias (the left-hand side figure) and its standard deviation (the right-hand side figure) for REDQ, REDQ+HER, and REDQ+HER+BQ. The results for all tasks are shown in Figs. 34 and 35.

---

[10]For a comprehensive investigation, REDQ+HER+BQ is also evaluated.

[11]We can observe value-estimation biases induced by HER only in a few tasks, e.g., FetchSlide-v1 and FetchPIckAndPlace-v1 (Figs. 34 and 35).

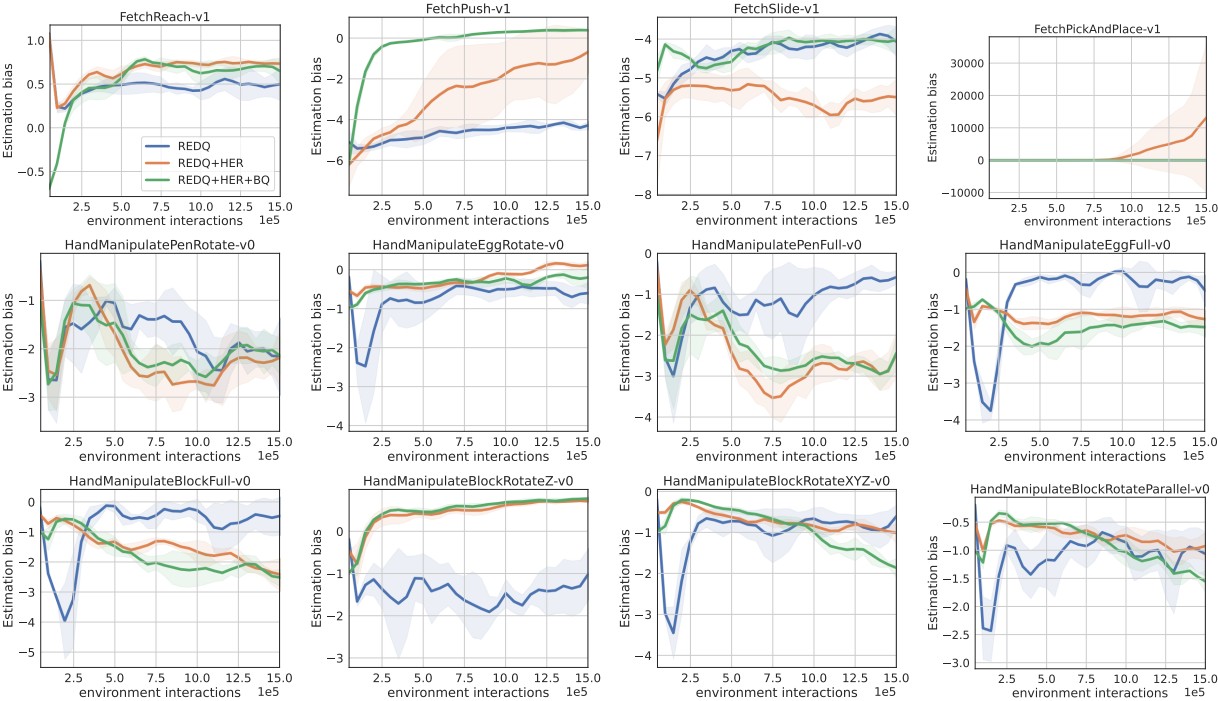

Figure 34: Estimation bias for REDQ, REDQ+HER, and REDQ+HER+BQ.

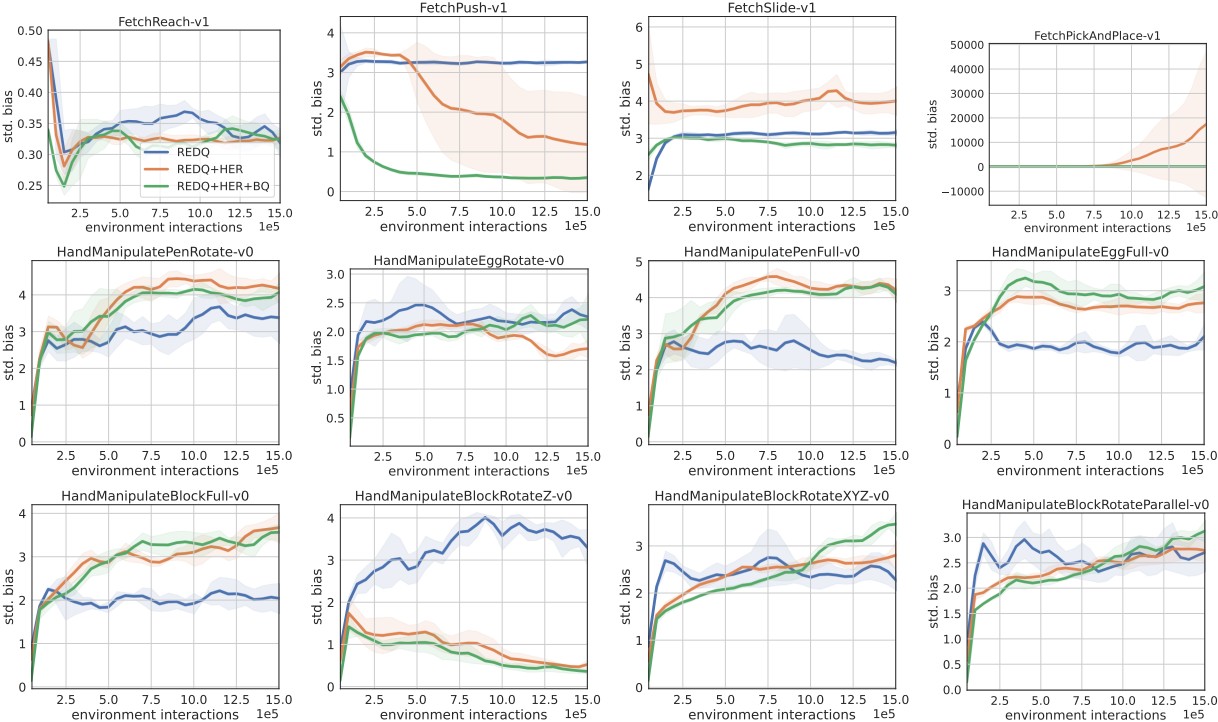

Figure 35: The standard deviation of estimation bias for REDQ, REDQ+HER, and REDQ+HER+BQ.

## D    Algorithmic Description of Reset (Nikishin et al., 2022) with Our Modifications

---

**Algorithm 2** Reset with our modifications (HER and BQ)

---

Initialize policy parameters $\theta$, two Q-function parameters $\phi_i$, empty replay buffer $\mathcal{D}$, and episode length $T$. Set target parameters $\bar{\phi}_i \leftarrow \phi_i$, for $i = 1, 2$.

1: Sample goal $g \sim p_g(\cdot)$ and initial state $s_0 \sim p_{s_0}(\cdot)$

2: **for** $t = 0, .., T$ **do**

3:     Take action $a_t \sim \pi_\theta(\cdot|s_t)$; Observe reward $r_t$ and next state $s_{t+1}$.

4:     **if** $t = T$ **then**

5:         $\mathcal{D} \leftarrow \mathcal{D} \bigcup \{(s_t, a_t, r_t, s_{t+1}, g)\}_{t=0}^T$; Select new goal $g_t'$; Calculate new reward $r_t' \leftarrow \mathcal{R}(s_t, a_t, g_t')$; $\mathcal{D} \leftarrow \mathcal{D} \bigcup \{(s_t, a_t, r_t', s_{t+1}, g_t')\}_{t=0}^T$

6:     **for** $G$ updates **do**

7:         Sample a mini-batch $\mathcal{B} = \{(s, a, r, s', g)\}$ from $\mathcal{D}$.

8:         Compute the target Q-value $y$:

$$y = r + \gamma \min \left( \max \left( \min_{i \in \{1,2\}} Q_{\bar{\phi}_i}(s', a', g), Q_{\min} \right), Q_{\max} \right) - \alpha \log \pi_\theta(a'|s', g), \quad a' \sim \pi_\theta(\cdot|s', g)$$

9:         **for** $i = 1, 2$ **do**

10:             Update $\phi_i$ with gradient descent using

$$\nabla_\phi \frac{1}{|B|} \sum_{(s,a,r,s',g) \in \mathcal{B}} (Q_{\phi_i}(s, a, g) - y)^2$$

11:             Update target networks with $\bar{\phi}_i \leftarrow \rho \bar{\phi}_i + (1-\rho)\phi_i$.

12:         Update $\theta$ with gradient ascent using

$$\nabla_\theta \frac{1}{|B|} \sum_{s \in \mathcal{B}} \left( \frac{1}{2} \sum_{i=1}^2 Q_{\phi_i}(s, a, g) - \alpha \log \pi_\theta(a|s, g) \right), \quad a \sim \pi_\theta(\cdot|s, g)$$

13:         **if** the number of environment interactions reaches a reset period **then**

14:             Reinitialize $\theta$ and $\phi_1, \phi_2$.

---

# E Hyperparameter Settings

Table 1: Hyperparameter settings

| Method | Parameter | Value |
|--------|-----------|-------|
| REDQ | optimizer | Adam (Kingma & Ba, 2015) |
| Reset | learning rate | $3 \cdot 10^{-4}$ |
| | discount rate $\gamma$ | 0.99 |
| | target-smoothing coefficient | 0.005 |
| | replay buffer size | $10^6$ |
| | number of hidden layers for all networks | 2 |
| | number of hidden units per layer | 256 |
| | mini-batch size | 256 |
| | random starting data | 10000 for HER-based methods and 5000 for the others |
| | replay ratio $G$ | 20 |
| | in-target minimization parameter $M$ | 2 |
| | ensemble size $N$ | 5 for REDQ and 2 for Reset. |
| HER | number of additional goals | 1 |
| BQ | upper bound of Q-value $Q_{\max}$ | 0 (i.e., $\sum_t^\infty \gamma^t \cdot 0$) |
| | lower bound of Q-value $Q_{\min}$ | -100 (i.e., $\sum_t^\infty \gamma^t \cdot -1 = \frac{-1}{1-\gamma}$ with $\gamma = 0.99$) |

