# OpenReview forum: "Efficient Sparse-Reward Goal-Conditioned Reinforcement Learning with a High Replay Ratio and Regularization"
_TMLR — Rejected by TMLR_

### Review · Reviewer_yyB8 · 2024-01-22

**Summary Of Contributions:**

The paper focuses on sparse reward goal conditioned MDP settings and shows how REDQ can be combined with hindsight experience replay to achieve state of the art performance on the gym Robotics environments.
Namely, the paper shows that naively combining REDQ with HER leads to unstable performance, which can be mitigated by clipping the targets for Q function to their maximal / minimal achievable values (which can be computed because the reward functions are bounded in these tasks).

**Audience:**

Yes

**Broader Impact Concerns:**

I do not see a broader impact concer.

**Claims And Evidence:**

Yes

**Requested Changes:**

* The reason for the HER instabilities and existing work on mitigating HER bias should be discussed.

* The paper mentions, that it distinguises from previous work by focusing on RL methods with high replay-ratio. However, it is not clear why the proposed techniques are specific to such methods. For example, it would be interesting to add REDQ+RR1 to the plots in Figure 10.

* Section 6 discusses several work that used bounded Q-values, but often in some different way. The paper should better discuss the different ways of Q value clipping.

* Instead of the "previous SOTA" curves, the plots should show the individual results of the different baselines, and if possible their learning curves instead of the final value.

* The definition of Sparse-Reward Goal Conditioned RL is a bit imprecise, as it lack the reward sparseness. The current definition would just as well apply to contextual RL.

* The writing could be improved as it often not accurate enough. The first paragraph distinguishes between "traditional methods" and "sample-efficient" methods, but looking at the citations, the traditional RL methods are even more recent. Maybe rather distinguish between on-policy and off-policy methods? Also the research questions are too imprecise: The answer to "Are both HER and BQ necessary to improve the performance of REDQ", is obviously "no", since there there are different techniques to improve REDQ. Also Q2 should be framed in a quantifiable way.

**Strengths And Weaknesses:**

Weaknesses:
* The contributions are quite limited. It is well-known that HER is an effective strategy for sparse reward goal-conditioned tasks. It is also well known that HER is biased, which can result in unstable optimization, e.g. see [1,2,3]. While I am not aware that Q-value clipping was used in this context (besides clipping between multiple critics), several related clipping techniques have been used in practice to stabilize RL. The paper already mentions several related work that performed some form of clipping, however, the exact same clipping technique (clipping Q values to 1/(1-gamma) r_max, or r_min respectively) was already done in an imitation learning setting [4].

* The results are purely empirical. The paper does not even mention that HER introduces a bias that might cause the instabilities.

* Given that the main contribution seems to be, that HER can be stabilized on some bounded reward tasks by Q-value clipping, I think that it would be important to compare this approach to existing techniques that try to address HER bias. Furthermore, experiments on different tasks would be very helpful to better understand the effect of Q-value clipping on HER stability.

Strengths:
* The empirical results demonstrate a good performance of the considered tasks.

---

> ### Author Response · Authors · 2024-02-12
> **Reply**
>
> Thank you for your valuable comments.
> We have uploaded a revised version of our manuscript.
> Major changes are highlighted in blue.
>
>
> **Q1:**
> It is also well known that HER is biased, which can result in unstable optimization, e.g. see [1,2,3]. ... however, the exact same clipping technique (clipping Q values to 1/(1-gamma) r_max, or r_min respectively) was already done in an imitation learning setting [4].
> **Q1':**
> The reason for the HER instabilities and existing work on mitigating HER bias should be discussed.
>
> **A1:**
> Could you provide the titles of references [1,2,3,4] (particularly references [1,2,3] discussing the bias of HER)?
> We will add a discussion about these papers and then include a discussion on the instability and bias of HER.
>
> **Q2:**
> The paper mentions, that it distinguises from previous work by focusing on RL methods with high replay-ratio. However, it is not clear why the proposed techniques are specific to such methods. For example, it would be interesting to add REDQ+RR1 to the plots in Figure 10.
>
> **A2:**
> The proposed technique (i.e., HER+BQ) has already been employed in low replay-ratio settings (DDPG) (cf. discussion with Reviewer xqQf).
> This is one of the reasons why we focus on methods (REDQ, or Reset) suited for a high replay-ratio setting.
> We revised the paper to clarify this point further.
>
> **Q3:**
> Section 6 discusses several work that used bounded Q-values, but often in some different way. The paper should better discuss the different ways of Q value clipping.
>
> **A3:**
> We used the method that bounds the target Q-value based on worst- and best-case Q-values.
> Other than this method, there are, for example:
> (i) the method that bounds the Q-function by employing an additional loss for Q-function learning,
> and (ii) the method that uses the empirical returns as the worst-case Q-value when bounding target Q-values.
> We added experiments on these methods to Appendix B.1 and B.2.
>
> **Q4:**
> Instead of the "previous SOTA" curves, the plots should show the individual results of the different baselines, and if possible their learning curves instead of the final value.
>
> **A4:**
> We have added plots of individual results to Appendix A.2.
> We retain figures with the previous SOTA in the main text, as incorporating scores for all five previous methods into one figure would reduce clarity.
>
> We plan to rerun experiments to plot individual learning curves for the previous methods, if conditions (especially on computational resources) permit.
>
> **Q5:**
> The definition of Sparse-Reward Goal Conditioned RL is a bit imprecise, as it lack the reward sparseness. The current definition would just as well apply to contextual RL.
>
> **A5:**
> We have made the following change to incorporate reward sparseness into the definition.
>
> Section 2, first paragraph:
> This is typically modeled as goal-augmented Markov decision processes $\langle \mathcal{S}, \mathcal{A}, \mathcal{G}, \gamma, p_{s_0}, p_g,  \mathcal{T}, \mathcal{R} \rangle$.
> ->
> This is typically modeled as goal-augmented Markov decision processes $\langle \mathcal{S}, \mathcal{A}, \mathcal{G}, \gamma, p_{s_0}, p_g,  \mathcal{T}, \mathcal{R} \rangle$ with sparse rewards.
>
>
> **Q6:**
> The writing could be improved as it often not accurate enough. The first paragraph distinguishes between "traditional methods" and "sample-efficient" methods...
>
> **A6:**
> Thank you for pointing out. We have made the following corrections.
>
> \> The first paragraph:
> As you pointed out, the use of 'traditional' induces a temporal inconsistency of the references, so we have removed it.
>
> \> Also the research questions are too imprecise: The answer to "Are both HER and BQ necessary to improve the performance of REDQ", is obviously "no", since there are different techniques to improve REDQ. Also Q2 should be framed in a quantifiable way.
>
> Are both HER and BQ necessary to improve the performance of REDQ?
> ->
> Does introducing both HER and BQ enhance REDQ's performance more than introducing HER or BQ individually?
>
> Does REDQ with HER and BQ perform as well as or better than previous SoTA methods?
> ->
> Does REDQ with HER and BQ achieve equal or superior performance compared to previous SoTA methods?

---

> > ### Comment · Reviewer_yyB8 · 2024-02-12
> > **missing citations**
> >
> > Thank you for your reply.
> > Sorry, I actually forgot to append the citations to my original review. Here they are:
> >
> > [1] Yang, R., Lyu, J., Yang, Y., Yan, J., Luo, F., Luo, D., ... & Li, X. (2021). Bias-reduced Multi-step Hindsight Experience Replay for Efficient Multi-goal Reinforcement Learning. arXiv preprint arXiv:2102.12962.
> >
> > [2] Liam Schramm, Yunfu Deng, Edgar Granados, & Abdeslam Boularias (2022). USHER: Unbiased Sampling for Hindsight Experience Replay. In 6th Annual Conference on Robot Learning.
> >
> > [3] Lanka, S., & Wu, T. (2018). Archer: Aggressive rewards to counter bias in hindsight experience replay. arXiv preprint arXiv:1809.02070.
> >
> > [4] Al-Hafez, F., Tateo, D., Arenz, O., Zhao, G., & Peters, J. (2022). LS-IQ: Implicit Reward Regularization for Inverse Reinforcement Learning. In The Eleventh International Conference on Learning Representations.

---

> > > ### Author Response · Authors · 2024-02-13
> > > **Reply to "missing citations"**
> > >
> > > Thank you for the prompt response.
> > > We will add a discussion of the references.

---

> > > ### Author Response · Authors · 2024-02-16
> > > **Additional revision**
> > >
> > > We uploaded the revised version of our paper.
> > >
> > > In the revision, we added a discussion about [1,2,3] to Section 6.
> > > Also, we added the experimental results for investigating HER bias in our tasks to Appendix C.

---

### Review · Reviewer_xqQf · 2024-01-25

**Summary Of Contributions:**

This paper combines REDQ, a RL algo that uses an high Replay Ratio (RR) and Regularisation for sample efficiency, and HER, that learns goal-reaching behaviour from sparse reward signals.

They observe that this combination leads to divergence of the Q function (mostly because of the bootstrapping target in REDQ and repeated goals from HER with an high RR). They simply solve this effect by clipping the Q-function between its maximal and minimal possible values (the call this trick BQ, for Bounded Q).

This results in a REDQ+HER+BQ training procedure, that achieves new SoTA performances on the Robotics tasks, and significantly outperforms the previous SoTA in average. Being not aware of the recent SoTA in sparse-reward robotics, I trust the authors on this comparison and let this jugement to the other reviewers.

**Audience:**

Yes

**Broader Impact Concerns:**

No ethical implications.

**Claims And Evidence:**

Yes

**Requested Changes:**

Please explain if I am misunderstanding the difference between your BQ and the target clipping from HER.
If I am not, the paper should be significantly changed to specify that this is not a new contribution but part of combining with HER.

**Strengths And Weaknesses:**

Strengths: The idea is novel, simple and well explained, while it achieve strong performances on robotic goal reaching tasks.

Weaknesses: The contribution is quite low: expanding one of the SoTA RL algorithm to sparse reward, using the most known method for sparse reward (HER).

Clipping the target (BQ) is already part of HER: "Moreover, we clip the targets used to train the critic to the range of possible values, i.e. [−1/(1−γ), 0]". (HER paper, appendix A, training procedure). So if I am not misunderstanding some detail, this should not be considered as part of the paper's contribution.

Details:
Section 3.2 (last paragraph): I'm not understanding the sentence about g'_t: by " the achieved
goal that comes from the same trajectories as the transition and was observed after it" do you simply mean s_T ?

Fig 4: because of the blue curve's variance (REDQ alone), it is very difficult to see the variance of the green curve. Here, maybe just keeping the orange and the green curves is sufficient.

---

> ### Author Response · Authors · 2024-02-12
> **Reply**
>
> Thank you for your valuable comments.
> We have uploaded a revised version of our manuscript.
> Major changes are highlighted in blue.
>
> **Q1:**
> Details: Section 3.2 (last paragraph): I'm not understanding the sentence about g'_t: by " the achieved goal that comes from the same trajectories as the transition and was observed after it" do you simply mean s_T ?
>
> **A1:**
> We meant the state randomly selected from $s_{t+1}, ..., s_{T}$. We have modified the description to make it clearer.
>
> **Q2:**
> Fig 4: because of the blue curve's variance (REDQ alone), it is very difficult to see the variance of the green curve. Here, maybe just keeping the orange and the green curves is sufficient.
>
> **A2:**
> We have replaced Fig. 4 with a new figure drawn with a clearer color scheme and additional textures.
> Is the revised Fig. 4 clear enough?
> If there are no issues with the revised figure, we will similarly update the other figures related to Q-value divergence (e.g., Figures 8 and 12).
> If the revised figure is worse than the previous one, we will revert to the previous figure and remove the blue curve as you recommended.
>
>
> **Q3:**
> Clipping the target (BQ) is already part of HER: "Moreover, we clip the targets used to train the critic to the range of possible values, i.e. [−1/(1−γ), 0]". (HER paper, appendix A, training procedure). So if I am not misunderstanding some detail, this should not be considered as part of the paper's contribution.
> **Q3'** Please explain if I am misunderstanding the difference between your BQ and the target clipping from HER. If I am not, the paper should be significantly changed to specify that this is not a new contribution but part of combining with HER.
>
> **A3:**
> Your understanding is correct.
> The original HER work and its subsequent works (e.g., HEREBP and VCP) applied the same approach of target clipping to the core RL method (DDPG).
>
> However, these previous works did not sufficiently clarify the properties of clipping:
> (i) They did not evaluate the extent of its contribution to performance improvements, nor its synergistic effects with HER. (Previous works conducted no ablation studies for clipping.)
> (ii) They did not sufficiently explain the rationale behind its use.
> (iii) They did not evaluate its effectiveness for RL methods other than DDPG.
>
> In response to these points, we made the following contributions:
> Regarding (i), we conducted explicit ablation studies for clipping and revealed its performance enhancements when used with HER (Figures 5 and 11).
> Regarding (ii), we experimentally demonstrated the rationale for using clipping, from the perspective of Q-function stability (Figures 4 and 12).
> Regarding (iii), we showed its effectiveness for REDQ and Reset (Figures 5 and 11).
> (These contributions would perhaps be beneficial for practitioners when they design or implement RL methods.)
>
> Based on the above, we have revised our paper as follows:
> (i) Clearly explain that clipping has been used in previous works on HER, and revise any parts that might suggest to the reader that BQ is our novel proposal.
> (ii) Add the aforementioned new contributions to the paper (e.g., replace Contribution 1 in the last paragraph of Section 1 with these new contributions).
> We have made major revisions, especially on Contribution 1 in Section 1, Section 3.1, and the third paragraph of Section 6.

---

> > ### Comment · Reviewer_xqQf · 2024-02-14
> > **Thank you for the clarifications**
> >
> > With the revisions, I think the paper is more clear about its contributions.
> > However, I am still on the fence regarding the impact of this work: it shows that when branching HER to REDQ, bounding the Q-value has an crucial importance. I think that is worth to be communicated, and useful for anyone aiming at working on sparse-reward tasks. But this is still quite limited in term of novelty and contribution.

---

> > > ### Author Response · Authors · 2024-02-16
> > > **Reply to "Thank you for the clarifications"**
> > >
> > > Thank you for your reply.
> > >
> > > We made several contributions other than bounding Q-values for REDQ.
> > > Among these, our finding that "In standard benchmarks for HER (i.e., Robotics [1] tasks), the base RL is a bottleneck" is likely to be impactful.
> > > We have shown that simply combining HER with a good RL method (REDQ) surpasses the previous methods that incorporate various technical enhancements to the HER (Section 4).
> > > This underscores the importance of using a good base RL method.
> > > Many previous works on Robotics tasks and HER, including those published as recently as 2023 (e.g., [2]), have traditionally used DDPG for a base RL method.
> > > Our findings advocate for rethinking this tradition.
> > > (#We do not mean to criticize the previous works on HER. Improvements on HER framework are indeed also important.)
> > >
> > > [1] https://github.com/Farama-Foundation/Gymnasium-Robotics
> > > [2] https://www.jair.org/index.php/jair/article/view/14398

---

### Review · Reviewer_rvcE · 2024-02-06

**Summary Of Contributions:**

This paper aims to adapt sample-efficient RL algorithms with a high replay ratio and regularization to sparse-reward goal-conditioned tasks. To this end, the paper proposes a framework that builds upon Randomized Ensembled Double Q-Learning (REDQ), and equips it with hindsight experience replay (HER) and bounding target Q-values (BQ). The experiments on 12 robot arm manipulation tasks show that the proposed method outperforms the baselines. While the overall efforts are appreciated, I am concerned with the motivation of this work (why REDQ+HER+BQ, not others), the novelty, the limited scope of the experiments (only manipulation), the clarity (section 4 and section 5), etc. In sum, I do not believe this work is ready for publication in its current form.

**Audience:**

Yes

**Claims And Evidence:**

Yes

**Requested Changes:**

See the paper weaknesses and questions section above

**Strengths And Weaknesses:**

## Paper strengths and contributions

**Clarity**

The proposed method is clearly explained in the method section.

**Experimental results**

The authors compared the proposed method with baselines in 12 manipulation tasks, and the results show that the proposed method mostly outperforms the baselines.

## Paper weaknesses and questions

**Citation hyperref**

The links of citations are broken.

**Clarity**
- Overall, Section 4 and Section 5 are very difficult to follow. I cannot clearly understand what the authors try to convey after reading them multiple times.
- The legend of the figures showing the learning performance is only shown in the first subfigure. It is quite difficult to read when looking at the last subfigure. I suggest putting a shared legend on the top of the entire figure.

**Novelty & contribution**

The novelty, from both the views of theoretical analysis aspect and empirical contribution perspective, is limited. The main contribution is integrating hindsight experience replay (HER) and bounding target Q-values (BQ) into Randomized Ensembled Double Q-Learning (REDQ). Yet, this combination seems like an arbitrary choice to me.
- RL algorithm: Why basing on REDQ? Why not other RL algorithms? It would be more convincing if this is fully justified from both theoretical perspectives and back with empirical results. Also, I am not sure if REDQ, published in 2021, is still the state-of-the-art algorithm now. If not, why not incorporate HER and BQ into the current best algorithms?
- Techniques: Why only HER and BQ? Why not other techniques, e.g., those employed in rainbow ("Rainbow: Combining Improvements in Deep Reinforcement Learning")? Again, this work would be more valuable if it provided theoretical results and experiments that verify the effectiveness of these two techniques while the ineffectiveness of others.

**Dense-reward environments**

I wonder how the proposed algorithm would work in environments with dense rewards. While I am well aware that the authors target sparse-reward goal-conditioned tasks, it would be informative to discuss dense-reward tasks since oftentimes we may not know an environment's reward setup before deploying RL algorithms.

**Related work**

I believe discussing recent skill-based RL works, which aim to solve sparse-reward tasks by leveraging offline task-agnostic datasets, can make the related work section more comprehensive.
- "Accelerating Reinforcement Learning with Learned Skill Priors"
- "Parrot: Data-driven behavioral priors for reinforcement learning"
- "Skill-based Meta-Reinforcement Learning"
- "Keep doing what worked: Behavioral modelling priors for offline reinforcement learning"

**Experiment results**

It would be informative to provide some descriptions of the behaviors of learned policies or present some qualitative results so that the readers can understand how/why one method performs better than another. Also, most sample-efficient figures show the aggregated IQM results of 4/8/12 tasks. It isn't easy to understand how each method performs on individual tasks.

**Manipulation tasks only**

While the proposed methods and some baselines were evaluated on 12 tasks, which should be considered sufficient, they are all manipulation tasks. I am wondering how well the proposed method and the baselines can perform in some goal-directed navigation or location tasks, such as
- AntReach (locomotion+navigation) used in "Generalizable Imitation Learning from Observation via Inferring Goal Proximity", "Learning Diverse Options via InfoMax Termination Critic"
- Point maze navigation used in "D4RL: Datasets for Deep Data-Driven Reinforcement Learning", "Accelerating Reinforcement Learning with Learned Skill Priors"
- Minigrid-like 2D navigation: https://github.com/Farama-Foundation/Minigrid

---

> ### Author Response · Authors · 2024-02-12
> **Reply 1/2**
>
> Thank you for your valuable comments.
> We have uploaded a revised version of our manuscript.
> Major changes are highlighted in blue.
>
>
> **Q1:**
> The links of citations are broken.
>
> **A1:**
> We added links to papers from citations e.g., "(Andrychowicz et al., 2020)".
>
>
>
> **Q2:**
> Overall, Section 4 and Section 5 are very difficult to follow. I cannot clearly understand what the authors try to convey after reading them multiple times.
> The legend of the figures showing the learning performance is only shown in the first subfigure. It is quite difficult to read when looking at the last subfigure. I suggest putting a shared legend on the top of the entire figure.
>
> **A2:**
> We rewrote Sections 4 and 5 to improve clarity.
> Besides, I will modify the legend of the figures.
>
>
> **Q3:**
> RL algorithm: Why basing on REDQ? Why not other RL algorithms? It would be more convincing if this is fully justified from both theoretical perspectives and back with empirical results. Also, I am not sure if REDQ, published in 2021, is still the state-of-the-art algorithm now. If not, why not incorporate HER and BQ into the current best algorithms?
>
> **A3:**
> While REDQ was proposed in 2021, it is still one of the best (most sample-efficient) algorithms for continuous control tasks (see [1] for example).
> We modified Section 3.1 to make this point clearer.
>
> (Also, REDQ is a descendant of DDPG (DDPG -> SAC -> REDQ), which was commonly used as the core RL algorithm in previous works on HER (e.g., [2, 3, 4]). Therefore, REDQ is well-suited for investigating how advancements in the core algorithm affect HER.)
>
>
> [1] Philip J Ball, Laura Smith, Ilya Kostrikov, and Sergey Levine. Efficient online reinforcement learning with offline data. arXiv preprint arXiv:2302.02948, 2023
>
> [2] Marcin Andrychowicz, Filip Wolski, Alex Ray, Jonas Schneider, Rachel Fong, Peter Welinder, Bob McGrew, Josh Tobin, Pieter Abbeel, and Wojciech Zaremba. Hindsight experience replay. In Proc. NeurIPS, 2017
>
> [3] Rui Zhao and Volker Tresp. Energy-based hindsight experience prioritization. In Proc. CoRL, 2018.
>
> [4] Jiawei Xu, Shuxing Li, Rui Yang, Chun Yuan, and Lei Han. Efficient multi-goal reinforcement learning via value consistency prioritization. Journal of Artificial Intelligence Research, 77:355–376, 2023
>
>
> **Q4:**
> Techniques: Why only HER and BQ? Why not other techniques, e.g., those employed in rainbow ("Rainbow: Combining Improvements in Deep Reinforcement Learning")? Again, this work would be more valuable if it provided theoretical results and experiments that verify the effectiveness of these two techniques while the ineffectiveness of others.
>
> **A4:**
> We do not claim that REDQ+HER+BQ is all we need for solving sparse-reward goal-conditioned tasks (and, since there are so many other techniques, it is difficult for us to comprehensively test them).
> We have made revisions to make this point clearer.
>
> Regarding theoretical insights, though it is indeed important, it is difficult for us to provide meaningful theoretical insights.
> We have added this to our future work in Section 7.
>
>
> **Q5:**
> I wonder how the proposed algorithm would work in environments with dense rewards. While I am well aware that the authors target sparse-reward goal-conditioned tasks, it would be informative to discuss dense-reward tasks since oftentimes we may not know an environment's reward setup before deploying RL algorithms.
>
> **A5:**
> Thank you for your suggestion, we will include a discussion about it.
> (Dense reward is often unbounded, and we might need to modify the way of bounding Q-value (e.g., [5]).)
>
> [5] Scott Fujimoto, Wei-Di Chang, Edward J Smith, Shixiang Shane Gu, Doina Precup, and David Meger. For sale: State-action representation learning for deep reinforcement learning. arXiv preprint arXiv:2306.02451, 2023
>
> **Q6:**
> I believe discussing recent skill-based RL works, which aim to solve sparse-reward tasks by leveraging offline task-agnostic datasets, can make the related work section more comprehensive.
>
> **A6:**
> Thank you for providing the references.
> We have added a discussion about the works to Section 6.
>
>
> **Q7:**
> It would be informative to provide some descriptions of the behaviors of learned policies or present some qualitative results so that the readers can understand how/why one method performs better than another. Also, most sample-efficient figures show the aggregated IQM results of 4/8/12 tasks. It isn't easy to understand how each method performs on individual tasks.
>
> **A7:**
> We will include descriptions of the learned behaviours.
> (Videos demonstrating the learned behaviours are already included in the supplementary files.)
>
> Regarding the sample-efficient figures, results for each method and each task are shown in the appendix.

---

> > ### Author Response · Authors · 2024-02-12
> > **Reply 2/2**
> >
> > **Q8:**
> > While the proposed methods and some baselines were evaluated on 12 tasks, which should be considered sufficient, they are all manipulation tasks. I am wondering how well the proposed method and the baselines can perform in some goal-directed navigation or location tasks, such as ...
> >
> > **A8:**
> > Thank you for your suggestion.
> > We will conduct experiments for these tasks.
> >
> > **Q9:**
> > While the overall efforts are appreciated, I am concerned with the motivation of this work (why REDQ+HER+BQ, not others), the novelty, the limited scope of the experiments (only manipulation), the clarity (section 4 and section 5), etc. In sum, I do not believe this work is ready for publication in its current form.
> >
> > **A9:**
> > The novelty of the work is not a criterion for acceptance or rejection decisions (cf. https://jmlr.org/tmlr/acceptance-criteria.html).
> > #We are very grateful for your valuable comments from a range of perspectives, and will make our best effort to reflect the comments.

---

### Decision · Action_Editor_X7NR · 2024-03-04

**Recommendation:** Reject

**Comment:**

The paper presents a combination of two known RL algorithms with Q-value clipping and an application to robotic manipulation task.

The results are good and might be of interest to practitioners in RL and more specifically robotic applications within RL. These types of paper could be valuable to the community, but their impact would be limited if not presented to the right audience.

The significance of the contribution is hard to judge because it is quite a specific application to an already small subfield (RL) of the wider general machine learning audience of TMLR. Taking into consideration that all the reviewers commented on the limited insight, I suggest the authors take the feedback into consideration and consider resubmitting an improved paper to a venue with a smaller scope or with one more appropriate to the specific application.

**Audience:**

The audience of the journal was discussed after the official recommendations were made.

Two of the three reviewers decided that the significance of the work is not large enough to interest the audience of the journal. One of the reviewers changed their assessments from their original one after a discussion about the novelty criterion (which TMLR does not impose). The review argued that the paper does not surface generalizable insights but rather simply restated them. Another reviewer also chose "No" here for a similar reason, and the positive reviewer also noted the limited contribution to due the paper being a combination of existing works (and empirical tricks like clipping) applied to a quite specific environment.

**Claims And Evidence:**

The claims made in the paper are supported by the evidence.

Several reviewers had comments regarding clarity, novelty, presentation, and related work. The authors provided thorough responses and prompt paper updates fixing the issues pointed out by the reviewers.

In the official recommendations, all of the reviewers agreed that the claims support the evidence.